# IMPROVING ROBUSTNESS IN VISION TRANSFORMERS WITH NULLSPACE NOISE AUGMENTED FINETUNING

## ABSTRACT

Enhancing the robustness of deep learning models, particularly in the realm of vision transformers (ViTs), is crucial for their real-world deployment. In this work, we explore the robustness of vision transformer models through the lens of nullspace, a fundamental concept in linear algebra, to propose a fine-tuning method that improves model robustness under various input perturbations. Our investigation centers on whether a vision transformer can exhibit resilience to input variations akin to the nullspace property in linear mappings, implying that perturbations sampled from this nullspace do not influence the model's output when added to the input. We confirm this by demonstrating the existence of a non-trivial nullspace in vision transformers, primarily attributed to the patch embedding layer. Moreover, we extend this idea beyond the linear layers, showcasing the feasibility of learning a non-linear counterpart (approximate nullspace) to the traditional nullspace for vision transformers through optimization techniques. Based on these insights, we propose a fine-tuning approach employing approximate nullspace noise to bolster the robustness of ViT models. Remarkably, within just a single epoch of fine-tuning, our method effectively mitigates the adverse effects of distribution shifts and adversarial perturbations across a wide spectrum of scenarios.

## 1 INTRODUCTION

The field of computer vision has witnessed significant advances, with the emergence of Vision Transformers (ViTs) (Dosovitskiy et al., 2021) marking a notable milestone. Following this advancement, a series of architectural refinements have been explored (Ali et al., 2021; Li et al., 2022; Liu et al., 2021), paving the way for the development of vision foundation models (Kirillov et al., 2023; Zou et al., 2023) by scaling up both the model and dataset. Despite these strides, robustness continues to pose a pivotal concern for their practical deployment. The weak inductive bias in transformers, while aiding expressive power, can easily learn biases present in the data (Bai et al., 2021b; Fu et al., 2022; Qin et al., 2022), hence compromising their ability to maintain consistent and reliable predictions across diverse real-world scenarios (Shi et al., 2023; Mazurowski et al., 2023).

Various methods have emerged aiming to enhance the robustness of transformer-based models. Predominantly, these methods are model-agnostic, deploying techniques like data augmentation (Xiao et al., 2023; Esser et al., 2021; Steiner et al., 2022; Liu et al., 2022) and regularization (Chen et al., 2022b; Steiner et al., 2022; Chefer et al., 2022), aligning with the overarching methodology in robustness studies (Wang et al., 2022; Liu et al., 2023). For instance, (Xiao et al., 2023) augments training data by masking image patches based on the class activation map and refilling them with randomly sampled images. Chen et al. (2022b) utilized a sharpness-aware optimizer to encourage a smooth loss landscape of the converged model. Despite the variety, a common theme among these approaches is their emphasis on external modifications or general optimization methods, overlooking the intrinsic properties of the models under consideration. This observation raises the following question: Can we leverage the inherent properties of ViTs to enhance their robustness?

This paper explores the robustness of transformer models from an algebraic perspective, identifying a similarity between model invariance to input perturbations and the concept of nullspace in linear algebra. We analogously define the (approximate) nullspace of transformer models to dissect their robustness behavior. The nullspace, a fundamental concept in linear algebra, refers to the subspace of a domain mapped to zero by a linear mapping. By definition, it is closed under addition and scalar

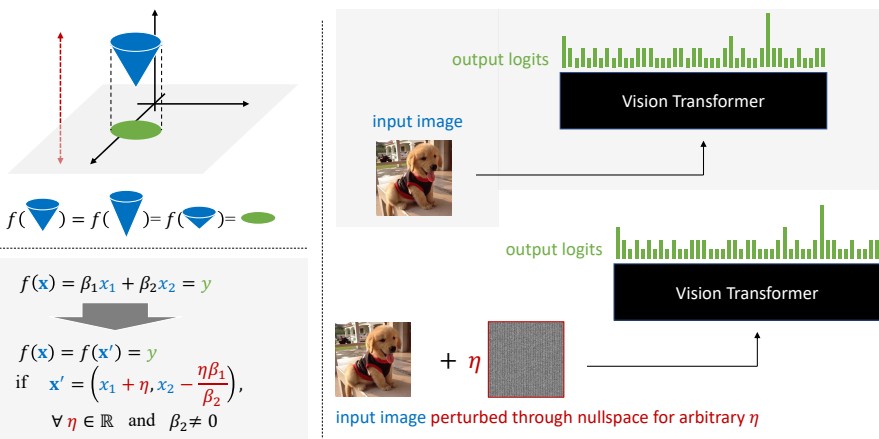

Figure 1: **An illustration of the nullspace in three cases (projection function case, left top; linear function case, left bottom; vision transformer case, right).** For the functions in these three cases, there exists some nullspace, and the output of the function with respect to the input will remain the same no matter how much perturbation is introduced to the input along the nullspace. Also, the nullspace is function-specific (model-specific) and will not vary for different samples.

multiplication, implying that any vector from the nullspace, when added to the input of the linear mapping, does not alter its output. Mirroring this, for nonlinear functions like transformers, we aspire for their outputs to remain invariant to certain perturbations. Our analysis starts with the observation that the patch embedding layer of transformers possesses a non-trivial nullspace in most common configurations, as guaranteed by the Rank-Nullity Theorem. Further, at the nonlinear encoder level, we identify an approximate nullspace through optimization techniques, which adequately adhere to the linear space properties of closure under addition and scalar multiplication. We find that employing noise sampled from this approximate nullspace for data augmentation significantly enhances model robustness in various scenarios.

**Comparison with the Concept of Invariance** The robustness literature extensively explores the notion of invariance. Depending on the problem at hand, researchers have delved into the invariance of models to norm-bounded perturbations (Madry et al., 2018b), cross-domain distribution shifts (Ganin et al., 2016), image texture, color, and background information Hermann et al. (2020); Wang et al. (2019); Ge et al. (2021), among others. The model nullspace concept we explore also epitomizes a form of invariance but deviates from specific robustness concepts. It hinges on the presumption that a robust model should remain invariant to a set of (potentially large) perturbations, without dictating how humans should interpret these perturbations. Through an optimization algorithm, we learn these perturbations from the model, unveiling the existence of an approximate nullspace within the model by demonstrating their similar properties to vector space elements. Our empirical findings underline the connection between this nullspace and robustness across diverse scenarios.

The main contributions of our paper include:

- We find rich connections between the robustness of vision transformers to the algebraic notion of nullspace, providing fresh insights into the intrinsic properties impacting model robustness. Our claims are substantiated by experimental results showing that expanding the approximate nullspace effectively improves the model robustness.
- We conduct comprehensive analysis on the existence of nullspace within transformer models. We establish the existence of nullspace at the patch embedding layer, and empirically identify an approximate nullspace at the nonlinear encoder level of transformers by validating their algebraic properties.
- We propose an effective fine-tuning method exploiting the identified approximate nullspace for data augmentation, enhancing model robustness without architectural modifications, thus requiring only fine-tuning with minimal additional data. Our method is empirically validated across multiple benchmark datasets, showing significant robustness improvements against adversarial and out-of-distribution scenarios.

This perspective on robustness, based on the algebraic properties of transformers, offers insights into aspects of transformer models that haven't been fully explored, suggesting directions for enhancing the robustness and reliability of vision models. By examining the concept of nullspace, we provide a new approach to understanding and improving the robustness of transformer-based vision models.

## 2 NULLSPACE ANALYSIS

Given an input vector $\mathbf{x} \in \mathbb{R}^{1 \times p}$ and a weight matrix $\beta \in \mathbb{R}^{p \times k}$, we can construct a simple linear regression model as follows:

$$\mathbf{y} = \mathbf{x}\beta \tag{1}$$

where $\mathbf{y} \in \mathbb{R}^{1 \times k}$ is the response vector. Intriguingly, for a given matrix $\beta$, there exists a set of vectors $\mathcal{N}$ such that

$$\mathbf{u}\beta = \mathbf{0}, \ \forall \mathbf{u} \in \mathcal{N} \tag{2}$$

In this case, for any given vector $\mathbf{x}$, the output for $\mathbf{x}$ and $\mathbf{x} + \mathbf{u}$ will be identical for any $\mathbf{u} \in \mathcal{N}$. Here, $\mathcal{N}$ is referred to as the nullspace of $\beta$. In the appendix (Section A.1), we elaborate further on the properties and interpretations of the nullspace.

### 2.1 VISION TRANSFORMER: RECAP

We first review the working mechanism of a vision transformer. Following it, we demonstrate the existence of nullspace for ViTs.

Vision transformer as introduced by Dosovitskiy et al. (2021) is a function $f_\omega$ with $\omega$ as the trainable weights. The function takes as input an image $\mathbf{x} \in \mathcal{X}^{c \times h \times w}$ and outputs a classification response $\mathbf{y} \in \mathcal{Y}^k$ over $k$ categories. $c$ is the number of channels (typically 3 for red, green, and blue), $h$, $w$ correspond to height and width of the input image. This neural network function can be broken down into 3 stages, namely:

- *patch embedding stage*, $f_\theta : \mathcal{X}^{c \times r \times r} \to \mathcal{U}^d$. This steps projects the input image patch of predetermined dimensions $c$, $r$ and $r$ to a one-dimensional embedding of length $d$. It is ensured that patches have no overlaps between them. Hence, the number of such non-overlapping patches generated from the input image are $m = \frac{h \times w}{r^2}$.
- *self-attention stage*, $f_\phi : \mathcal{U}^{(m+1) \times d} \to \mathcal{V}^{(m+1) \times d}$. In the next step, the generated patch embeddings are passed through layers of self-attention modules to process long range interactions amongst them. Apart from the $m$ patch embeddings an additional embedding in form of `cls` token is utilised in this step.
- *classification stage*, $f_\psi : \mathcal{V}^d \to \mathcal{Y}^k$. The final step is to perform the $k$-way classification. For this, we simply keep the processed encoding corresponding to `cls` token and project it through a linear classification layer.

This breakdown can also be illustrated as:

$$\mathbf{x}_i^{c \times r \times r} \underset{f_\theta}{\longrightarrow} \mathbf{u}_i \qquad [\mathbf{u}_{cls}\mathbf{u}_0 \dots \mathbf{u}_m] \underset{f_\phi}{\longrightarrow} [\mathbf{v}_{cls}\mathbf{v}_0 \dots \mathbf{v}_m] \qquad \mathbf{v}_{cls} \underset{f_\psi}{\longrightarrow} [y_0 y_1 \cdots y_k] \tag{3}$$

### 2.2 NULLSPACE OF PATCH EMBEDDING

The nullspace of a vision transformer comes from the fact that its first layer is a linear transformation layer. As per the rank-nullity theorem[1], a non-trivial nullspace of the patch embedding layer always exists if $cr^2 > d$. In practice for many ViT based architectures, we find that this is the case (results reported in Section A.4).

Now considering the ViT models where a nullspace exists in the patch embedding layer, and let $\theta$ denote the weight matrix. Then for an input patch $\mathbf{x} \in \mathbb{R}^{c \times r \times r}$, we have the representation (embeddings) learned from that patch as

$$\mathbf{y} = \mathbf{x}\theta.$$

---

[1] Rank-nullity theorem states that the sum of the rank of a matrix and dimension of its nullspace should be equal to the number of columns of the said matrix.

Given a matrix, finding its nullspace is a standard practice[2]. Let $B_\theta = \{\mathbf{b}_1, \mathbf{b}_2, \ldots \mathbf{b}_k\}$ be the $k$ basis vectors for this nullspace. As per the axioms of nullspace (A.1), we can sample an element from $\mathcal{N}_\theta$ as:

$$\mathbf{v} = \lambda_1 \mathbf{b}_1 + \lambda_2 \mathbf{b}_2 + \cdots + \lambda_d \mathbf{b}_k. \qquad (4)$$

The property of such a sample will be that the output of the patch embedding will effectively remain preserved, $f_\theta(\mathbf{x} + \mathbf{v}) = f_\theta(\mathbf{x})$. Since the output after the first layer remains unaffected the final output of the classification remains unchanged. In this manner, nullspace of patch embedding also serves as a subset of the nullspace of the vision transformer.

We refer to $\mathbf{v}$ as **nullspace noise** or **nullspace perturbation**. It is important to note that nullspace noise only depends on the patch embeddings weights and is independent of the input. As a result, the same noise can be added to any input without impacting the final output.

## 2.3 THE ENCODER-LEVEL NULLSPACE

So far we have demonstrated that a non-trivial nullspace exists for the patch embedding layer, and hence the entire vision transformer is invariant to all perturbations in that space. In this section we move further down the structure of ViT and investigate whether the encoder is also invariant to certain perturbations. To recall, self-attention stage applies a series of *QKV* attention operations followed by normalization and non-linear transformations. The overall operation is thus non-linear, which means the notion of nullspace cannot be directly applied to $f_\phi$. Regardless, we can still attempt to preserve the axiom of most interest to us, *closeness under vector addition*. We attempt this through considering the property of certain noise vectors that, when added to any input, does not disturb the output of a function. Therefore, instead of looking for a vector space, we can instead search for a set with the following property:

$$\tilde{\mathcal{N}}_\phi = \{\mathbf{v} | f(\mathbf{u} + \mathbf{v}) = f(\mathbf{u}) \; \forall \mathbf{u} \in \mathcal{U}\}. \qquad (5)$$

i.e. adding elements from $\tilde{\mathcal{N}}_\phi$ [3] to the input of $f_\phi$ has no impact on the output. For the patch embedding layer $f_\theta$ discussed in Section 2.2 , it is easy to verify that any vector sampled from its nullspace $\mathcal{N}_\theta$ satisfies this property and thus belongs to $\tilde{\mathcal{N}}_\theta$. To study this property in nonlinear functions, we extend the definition of nullspace in linear algebra, and refer to $\tilde{\mathcal{N}}_\phi$ as the *encoder-level nullspace* of transformer. If such a space exists, it directly implies that the transformer model is robust to certain perturbations in the input space. Our theoretical analysis established the following sufficient conditions for the existence of a nontrivial encoder-level nullspace. (The complete proof is given in Appendix A.)

**Proposition 1.** *Consider a self-attention layer with $h$ heads and $\{(\mathbf{Q}_i, \mathbf{K}_i, \mathbf{V}_i)\}_{i=1}^h$ as its query, key and value projection matrices. If the following conditions are met*

1. *$\mathbf{Q}_i \mathbf{K}_i^\top$ is symmetric for $i = 1, \ldots, h$*

2. *The row space $\mathrm{R}(\mathbf{V}_i^\top) \subseteq \mathrm{R}(\mathbf{Q}_i \mathbf{K}_i^\top)$ for $i = 1, \ldots, h$*

3. *for some $m \neq n$, $\mathbf{Q}_m \mathbf{K}_m^\top$ has colinearity with $\mathbf{Q}_n \mathbf{K}_n^\top$, i.e. for some $k$ the $k$th row of $\mathbf{Q}_m \mathbf{K}_m^\top$, denoted as $\mathbf{r}_{m,k}$, satisfies $\mathbf{r}_{m,k} \neq \mathbf{0}$ and $\mathbf{r}_{m,k} \in \mathrm{R}(\mathbf{Q}_n \mathbf{K}_n^\top)$*

*then there exists at least one $\mathbf{W}$ such that $\mathbf{W} \neq \mathbf{0}$ and $\mathrm{head}_i(\mathbf{X} + \mathbf{W}) = \mathrm{head}_i(\mathbf{X})$ for all attention head $i$ in this layer and arbitrary $\mathbf{X}$.*

**Remark.** *Condition 1 can be met if $\mathbf{Q}_i$ and $\mathbf{K}_i$ satisfy some special relation. For example, let $\mathbf{P}\mathbf{D}\mathbf{P}^{-1}$ be a diagonalization of a real symmetric matrix $\mathbf{A}$. If $\mathbf{Q}_i = \mathbf{B}\mathbf{P}$ and $\mathbf{K}_i = \mathbf{B}(\mathbf{P}^{-1})^\top \mathbf{D}$, then we have $\mathbf{Q}_i \mathbf{K}_i^\top = \mathbf{B}\mathbf{A}\mathbf{B}^\top$ to be symmetric.*

*In addition, evidence has shown that, $\mathbf{Q}_i \mathbf{K}_i^\top$ can be empirically symmetric, especially for ViTs, when the attention heads are visualized and correlation of parameters is calculated (Yeh et al., 2023)*

Although our theory suggests a sufficient condition for the existence of encoder-level nullspace, analytically finding $\tilde{\mathcal{N}}_\phi$ or probing its existence for generic transformers is challenging. Thus, as an

---

[2]Given a matrix, there are well established algorithms to find its nullspace. We refer the readers to detailed discussions such as (Kwak & Hong, 2004; Strang, 2009b;a). In our work, we rely on the Python package of *Numpy* for finding the nullspace of any given matrix (Harris et al., 2020).

[3]We use the tilde accent ~ to distinguish our extended definition $\tilde{\mathcal{N}}$ from the nullspace $\mathcal{N}$ in linear algebra.

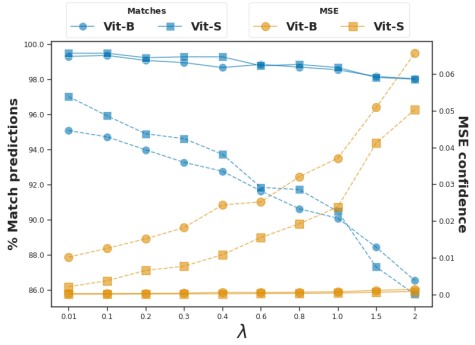
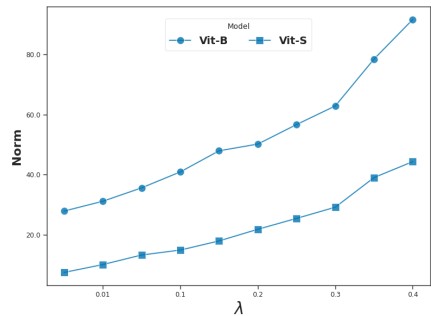

(a) Noise influence on the model output under different regularization strengths

(b) $\ell_2$ norm of learned noise under different regularization strengths

Figure 2: Exploratory experiments on the encoder-level nullspace. (a) Solid lines (–) represents the model performance under the learned noise, and dashed lines ($\cdots$) represent the performance after permutation of the learned noise. (b) by changing the regularization strengths, we explore noise in the encoder-level nullspace at different magnitudes.

exploratory experiment, we employ a numeric method: we search for individual element of this set, $\tilde{\mathbf{v}}_\phi$, an additive perturbation that brings minimal influence to the output of $f_\phi$ on the data distribution. We introduce a regularization term on the norm of $\tilde{\mathbf{v}}_\phi$ to prevent the trivial solution of $\mathbf{0}$.

$$\mathcal{L}_\phi(\tilde{\mathbf{v}}) = \mathbb{E}_{\mathbf{u} \in \mathcal{D}} \| f_\psi(f_\phi^0(\mathbf{u} + \tilde{\mathbf{v}})) - f_\psi(f_\phi^0(\mathbf{u})) \| - \lambda \log(\|\tilde{\mathbf{v}}\|) \qquad (6)$$

where $\|\cdot\|$ is the $\ell_2$ norm, $f_\phi^0$ is the representation of the `cls` token output by $f_\phi$, and $\lambda$ is the regularization coefficient. We find that simple gradient descent with Equation 6 works well in practice. Note that $\mathcal{L}_\phi(\tilde{\mathbf{v}})$ could potentially be unbounded below as $\|\mathbf{v}\| \to \infty$ due to the second term. Future work may prevent the trivial solution by using a lower bound on the noise norm for constrained optimization.

Equation 6 minimizes the $\ell_2$ norm between the predicted logits with and without the noise. Alongside the self-attention stage, we have also incorporated the classification stage into the loss, since the target of our study is to minimize the impact on the final output of the network. To learn the noise vector, we initialize $\tilde{\mathbf{v}}$ by sampling from a uniform distribution, and minimized the loss with gradient descent. We used ViT-S and ViT-B models with patch size 32 for evaluation. We used ImageNette (Howard, 2018) as the dataset for this experiment, which is a subset of ImageNet consisting of 10 categories. We learnt $\tilde{\mathbf{v}}$ on the training dataset ($\approx 9500$ images) and performed evaluation on the validation set ($\approx 4000$ images).

To quantitatively evaluate learned $\tilde{\mathbf{v}}_\phi$, in Fig. 2 (a) we report the percentage of matching classifications with and without learned nullspace noise, and the mean squared error computed at the output probabilities (hereafter "MSE probability"). We consider a prediction to be matched if the assigned category for input is the same with and without adding the perturbation. By varying the regularization strength, we get noise vectors of different magnitude (Fig. 2 (b)), all being fairly benign to the model's predictions. However, if we randomly reset the vector' direction by permuting their elements, the noise significant degrades the model's predictions. The experiment shows the feasibility of learning elements that approximately conform to our above definition of encoder-level nullspace with pretty good precision, and also indicates that at different magitudes there are certain directions in the input space toward which the perturbation is fairly benign to the model. To further show that those individually learned noise vectors are not optimization artifacts, but they reflect some intrinsic properties of the transformer model, we further did experiments to evaluate their behaviors under combination, and find remarkable similarity between their properties and some axioms of vector space, i.e. closure under addition and scalar multiplication. For detailed results and discussion see A.5

## 3 NULLSPACE NOISE AUGMENTED FINETUNING

In the previous section we demonstrated that there may exist a non-isotropic space in the input space of the vision transformer, which consists of perturbations in certain directions that the model is

insensitive to. As an intuitive explanation, there exits additive noise which is large enough to be human-perceptible but does not change the semantic features of images, and therefore should not change the prediction of a robust model (one such example is the Gaussian noise). In our previous formulation, we tried to find such noises specific to each model, and experiments results indicate that although it is difficult to find them perfectly (without any perturbation to the model output), we can effectively learn them if a certain degree of error is allowed. To more accurately quantify this space. we can choose a small $\epsilon$ and define $\epsilon$-approximate nullspace as follows

$$\tilde{\mathcal{N}}_\phi(\epsilon) = \{\tilde{\mathbf{v}} | \mathbb{E}_{\mathbf{u} \in \mathcal{D}} \|f(\mathbf{u} + \tilde{\mathbf{v}}) - f(\mathbf{u})\| \le \epsilon\}. \tag{7}$$

where $f(\cdot) = \text{Softmax}(f_\psi(f_\phi^0(\cdot)))$. It is easy to verify that $\forall \epsilon > 0, \mathbf{0} \in \tilde{\mathcal{N}}_\phi(\epsilon)$, and that $\forall \epsilon_2 > \epsilon_1 > 0, \tilde{\mathcal{N}}_\phi(\epsilon_1) \subseteq \tilde{\mathcal{N}}_\phi(\epsilon_2)$. Among the elements in $\tilde{\mathcal{N}}_\phi(\epsilon)$, we are more interested in those near the boundary, because they are characteristic of the value of $\epsilon$, and may reflect the size of $\tilde{\mathcal{N}}_\phi(\epsilon)$.

We hypothesize that the model's tolerance to approximate nullspace noise is indicative of its robustness under a variety of distribution shifts. Therefore, data augmentation with the learned noise is likely to boost the model's robustness. To test this hypothesis, we employ a bi-level optimization approach, where the inner problem finds the best noise vector according to Equation 9, and the outer problem finds the model that is the most tolerant to such noise, as showed in Equation 8.

$$\min_\phi \quad \mathbb{E}_{\mathbf{u} \in \mathcal{D}} \ell(f_\psi(f_\phi^0(\mathbf{u} + \tilde{\mathbf{v}}_\phi^*)), \mathbf{y}) \tag{8}$$

$$\text{where} \quad \tilde{\mathbf{v}}_\phi^* = \arg \max_{\tilde{\mathbf{v}}} \|\tilde{\mathbf{v}}\| \tag{9}$$

$$\text{s.t.} \quad \tilde{\mathbf{v}} \in \mathcal{N}_\phi(\epsilon)$$

where $\ell(\cdot)$ is the cross-entropy loss. While this optimization problem can also be solved in different ways, we used an efficient heuristic: we initialize the noise with a large enough sampling limit, minimize $\mathcal{L}_\phi(\tilde{\mathbf{v}})$ by gradient descent according to the loss function in Equation 10, and early stop it as soon as it enters $\tilde{\mathcal{N}}_\phi(\epsilon)$, as shown in Equation 11.

$$\mathcal{L}_\phi(\tilde{\mathbf{v}}) = \mathbb{E}_{\mathbf{u} \in \mathcal{D}} \|f_\psi(f_\phi^0(\mathbf{u} + \tilde{\mathbf{v}})) - f_\psi^0(f_\phi(\mathbf{u}))\| \tag{10}$$

$$\hat{\mathbf{v}}^* = \text{SGD}\left(\mathcal{L}_\phi(\tilde{\mathbf{v}}), \tilde{\mathbf{v}}_0, \epsilon\right) \tag{11}$$

Here, $\hat{\mathbf{v}}_\phi^*$ is the approximate solution for $\tilde{\mathbf{v}}_\phi^*$, $\text{SGD}(\mathcal{L}_\phi(\tilde{\mathbf{v}}), \tilde{\mathbf{v}}_0, \epsilon)$ denotes the gradient descent algorithm that minimizes the loss $\mathcal{L}_\phi(\tilde{\mathbf{v}})$ starting from its initial value $\tilde{\mathbf{v}}_0$ until it satisfies the condition $\mathcal{L}_\phi(\tilde{\mathbf{v}}) < \epsilon$. The noise norm starts from a large value and gets gradually reduced during the process. When early stopping is triggered, we obtain noise vectors that are close to the boundary of the $\epsilon$-approximate nullspace. For more details of our method, please refer to Algrothm 1 in Appendix A.3.

## 4 EXPERIMENTS

### 4.1 IMPLEMENTATION DETAILS

In this section, we conducted comprehensive evaluation of our nullspace augmentation method (Section 3) on several benchmarks. By making the model more tolerant to noise in the $\epsilon$-approximate nullspace, we hope to expand the nullspace itself and observe its effect on the model's robustness under different settings.

Starting from a pretrained model, we use the $\epsilon$-approximate nullspace noise as data augmentation to fine-tune the model. The noise is generated every 40 training steps according to Equation 11 with an $\epsilon$ of 0.03. The experiment was done within one epoch of training on the ImageNet-1k dataset. We used the vanilla `ViT-small` and `ViT-base` models, and `ViT-base(DAT)` which is the current SOTA on ImageNet-C dataset on the EasyRobust benchmark[4], trained using Discrete Adversarial Training proposed by Mao et al. (2022b). We evaluated the model performance in a wide range of settings to test its performance on the i.i.d dataset, under adversarial attacks and distribution shifts. For adversarial attacks we utilize FGSM (Goodfellow et al., 2015) and DamageNet (Chen et al., 2022a) as white-box and black-box attacks respectively. For distribution shift we employ

---

[4]https://github.com/alibaba/easyrobust

Table 1: **Effect of nullspace training on different models evaluated on multiple benchmark datasets.** Excluding DAT, vanilla ViT-S and ViT-B, the values for the baselines are directly reported from the corresponding papers. For DAT, we report the reproduced results following their evaluation setting.

| Methods | Clean | Adversarial Robustness | | | | | Out of Distribution Robustness | | | | | | Average |
| --- | --- | --- | --- | --- | --- | --- | --- | --- | --- | --- | --- | --- | --- |
| | | PatchFool | CW | MIM↓ | FGSM | DamageNet | A | C↓ | V2 | R | Sketch | Stylized | |
| ViT-S | 74.19 | 0.68 | 0.18 | – | 13.79 | 29.82 | 16.35 | 71.13 | 62.51 | 34.67 | 14.26 | 12.15 | 31.85 |
| ViT-S + NS (ours) | **77.47** | **19.10** | **2.38** | – | **25.95** | **32.43** | **20.77** | **55.98** | **66.5** | **41.61** | **25.67** | **16.02** | **38.94** |
| ViT-B | 77.68 | 15.92 | 0.56 | 81.72 | 25.65 | 38.69 | 23.88 | 62.16 | 66.05 | 41.63 | 16.31 | 17.97 | 38.41 |
| ViT-B + PR (Qin et al., 2022) | 78.20 | – | – | – | – | – | – | 47.60 | – | – | – | – | – |
| ViT-B + RandAugment + PR | 79.30 | – | – | – | – | – | – | 43.60 | – | – | – | – | – |
| ViT-B + AugMix + PR | 79.30 | – | – | – | – | – | – | **41.60** | – | – | – | – | – |
| RobustVit(Mao et al., 2022d) | 81.90 | – | – | – | 51.80 | – | 28.50 | 46.80 | – | 48.70 | 36.00 | – | – |
| ViT-B + PAT(Herrmann et al., 2022) | 81.71 | – | – | – | – | – | – | 44.99 | **70.82** | 47.66 | **36.77** | 19.14 | – |
| Discrete-ViT(Mao et al., 2022a) | 79.48 | – | – | – | 45.76 | **44.91** | 17.20 | 46.22 | 68.05 | 44.77 | 34.59 | **19.38** | **45.32** |
| AGAT (Wu et al., 2022) | 70.41 | – | **50.84** | – | – | – | – | – | – | – | – | – | – |
| Fan-ViT-B (Zhou et al., 2022) | **83.60** | – | – | – | – | – | **35.40** | 44.40 | – | 51.80 | – | – | – |
| TORA($\lambda = 0.9$) (Li & Xu, 2023) | 80.30 | – | – | – | 74.2 | – | 22.20 | **41.60** | – | **53.70** | – | – | – |
| Relevance Maps (Chefer et al., 2022) | 80.40 | – | – | – | – | – | 3.00 | – | 69.8 | 35.40 | 35.80 | – | – |
| ViT-B + NS(ours) | 81.42 | **23.52** | 2.36 | 74.72 | 36.50 | 40.44 | 24.55 | 47.82 | 70.25 | 44.85 | 26.35 | 19.02 | 43.95 |
| ViT-B + DAT(Mao et al., 2022b) | **81.47** | 22.64 | 0.76 | **70.28** | 48.80 | 43.31 | 23.83 | 45.95 | **70.24** | 48.68 | 36.94 | **23.99** | 47.92 |
| ViT-B + DAT + NS(ours) | 81.33 | **24.14** | **0.88** | 71.18 | **48.98** | **43.67** | **24.22** | **45.91** | 70.14 | 48.48 | **37.25** | 23.87 | **48.01** |

ImageNet-C (Hendrycks & Dietterich, 2018), ImageNet-A (Hendrycks et al., 2021b), ImageNet-V2 (Recht et al., 2019), Imagenet-R (Hendrycks et al., 2021a), ImageNet-Sketch (Wang et al., 2019) and Stylized-Imagenet (Geirhos et al., 2018). ImageNet-C consists of validation modified by applying corruptions in the form of weather effects, noises, etc. ImageNet-A applies the imagenet objects in hard contexts. ImageNet-R and ImageNet-Sketch consist of imagenet categories in different art forms. ImageNet-Stylized applies texture transfer onto the ImageNet validation images to create shape-texture contradictions. We used the EasyRobust library Mao et al. (2022c) for code implementation and the checkpoints of `ViT-base(DAT)`, and replicated their results. For more implementation details please see our supplementary document.

## 4.2    EXPERIMENT: ROBUSTNESS EVALUATION

We evaluate the effect of nullspace training for improving the robustness of vision transformers under different settings. We used the official mCE score as the evaluation metric for ImageNet-C, where lower mCE indicates better robustness, and used the accuracy score for all the other settings. We used $100 - \mathrm{mCE}$ before taking the average over all settings.

The result in Table 1 shows that our nullspace augmentation method consistently improves the robustness of models under distribution shifts and adversarial attacks, yielding a large gain in average performance for the vanilla `ViT-small` and `ViT-base model`, and slightly outperforms the current SOTA model. This not only shows that our nullspace training method is effective but also validates our previous hypothesis about the connection between the tolerance to nullspace and the robustness of transformer models.

## 4.3    EXPERIMENT: ADVERSARIAL FINETUNING

In this experiment, we compare our method with fine-tuning using two PGD adversarial training methods, Madry(Madry et al., 2018a) and TRADES(Zhang et al., 2019) on the ViT-S model. TRADES, in each training iteration, generates adversarial examples using PGD and updates the model's parameters to minimize the worst-case loss on these adversarial examples while also minimizing the standard classification loss on clean data. Madry, on the other hand, focuses exclusively on minimizing the worst-case loss on adversarial examples.

In Table 2, we observe that Madry and TRADES, they provide better performance for adversarial evaluation. This is expected as the methods are catered for improving adversarial robustness. However, this exclusivity leads to relatively poorer performance in a wider benchmark evaluation. Compared to our method, Madry and TRADES perform considerably lower in the natural

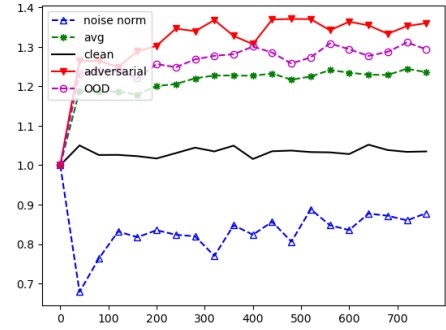

Figure 3: **Change trend of multiple metrics with training steps.** "Adversarial" is the average performance on "FGSM" and "DamageNet" settings, "OOD" is the average score of the six OOD datasets, and "avg" is the total average. All values are divided by their initial values to show the trend more clearly.

OOD setting. Lastly, Madry also leads to drop in performance when compared to the vanilla trained baseline for natural images.

Table 2: **Comparison of nullspace augmented fine-tuning with PGD based adversarial robustness methods of Madry and TRADES.** We report the performance for a ViT-S model.

| Method | clean | FGSM | DamageNet | A | C ($\downarrow$) | V2 | R | Sketch | Stylized |
|--------|-------|------|-----------|------|------|-------|-------|--------|----------|
| ViT-S | 74.19 | 13.79 | 29.82 | 16.35 | 71.13 | 62.51 | 34.67 | 14.26 | 12.15 |
| Madry | 70.53 | **39.37** | **49.91** | 9.37 | 81.74 | 58.88 | 39.04 | 21.36 | 10.76 |
| TRADES | 74.02 | 38.85 | 36.28 | 16.53 | 73.11 | 63.37 | 40.86 | 26.43 | 13.22 |
| Ours | **77.47** | 25.95 | 32.43 | **20.77** | **55.98** | **66.5** | **41.61** | **25.67** | **16.02** |

## 4.4 Training Progression

We track the $l_2$ norm of the learned noise and various performance metrics during the nullspace training. As shown in Fig. 3. Before the nullspace training, it was hard to optimize the noise into the $\epsilon$ region even with increased training , so the norm started with a high value. As the nullspace training started, we found from the experiment log that the noise was always able to enter the $\epsilon$ region. In Appendix A.6, we show the MSE probability of the learned noise vectors, which are smaller than $\epsilon$ at each round of noise learning. Also, the norm of the learned noise gradually increased along the process of model fine-tuning. The fluctuation may have mainly resulted from the randomness in mini-batches and the optimization dynamics. The model allows for noises with larger and larger norms to be within $\epsilon$-approximate, which informally suggests an increasing $\epsilon$-approximate nullspace. Accompanied by the trend is the increase in robustness scores in both OOD and adversarial settings, which further corroborates our findings.

## 4.5 Ablation Study

We conduct an extensive study to analyse the performance of our method under choice of $\epsilon$. Furthermore, we also compare our approach with a simple baseline of using an $\epsilon$ noise sampled from a Gaussian distribution.

From table 3, it can be inferred that the nullspace noise based finetuning is relatively robust to the choice of $\epsilon$. Moreover, compared to using randomly generated $\epsilon$-noise, our nullspace based training provides significant performance boost. This observation stands across different values of $\epsilon$.

Table 3: **Impact of $\epsilon$ on the final performance**. Moreover, we also compare our approach against random $\epsilon$ noise based finetuning.

| $\epsilon$ | Finetuning | FGSM | DamageNet | A | C ($\downarrow$) | V2 | R | Sketch | Stylized |
|------|-----------|-------|-----------|-------|-------|-------|-------|--------|----------|
| 0.01 | nullspace | 26.04 | 33.65 | 20.45 | 56.26 | 66.47 | 41.4 | 23.34 | 15.85 |
| | random | 21.54 | 28.81 | 17.07 | 55.13 | 61.98 | 34.97 | 14.43 | 12.14 |
| 0.03 | nullspace | 25.95 | 32.43 | 20.77 | 55.98 | 66.5 | 41.61 | 25.67 | 16.02 |
| | random | 23.18 | 29.61 | 16.91 | 54.68 | 62.2 | 35.05 | 14.77 | 12.34 |
| 0.1 | nullspace | 25.38 | 33.09 | 20.16 | 56.41 | 66.47 | 40.42 | 22.66 | 15.78 |
| | random | 23.93 | 30.56 | 16.47 | 54.52 | 62.48 | 34.66 | 14.99 | 12.35 |

## 5 Related Work

**Nullspace and Neural Networks** To the best of our knowledge, the earliest work investigating neural networks alongside nullspaces corresponds to the study by Goggin et al. (1992). They studied the universal approximation capabilities of a multi-layered perceptrons by comparing the outputs and nullspace of inputs. Through a classical example of *learning XOR* they showed that with the use of hidden layers, an MLP is able to construct a transformation which maps input to targets even if the target happens to be in the nullspace of the input. In a much recent work, Sonoda et al. (2021) mathematically specified the behavior of nullspace, but only for fully connected networks. On the application side, for a continual learning setting, Wang et al. (2021) proposed to map new tasks to the nullspace of the existing tasks. Lastly as a novel architecture, Abdelpakey & Shehata (2020) proposed NullSpaceNet which maps inputs from the same category to a joint-nullspace instead of a feature space.

**Invariance** Model invariance to (subjectively small) variations in the input is a well studied topic in deep learning research. Moreover so, relationship of being robust to input perturbations and

model generalisation is also widely accepted (Zhang et al., 2017; Rebuffi et al., 2021). For studying and understanding the invariance of models to input perturbations, the nature of perturbations is heuristically (or subjectively) selected based on the task at hand (Arjovsky et al., 2019; Chen et al., 2019; Cohen & Welling, 2016; Zhang, 2019; Satorras et al., 2021). Unlike prior studies, we do not impose any restrictions on the input noise apart from it resembling non-trivial nullspace noise. To the best of our knowledge, the nullspace perspective of invariance has not yet been explored.

**Robustness in ViT** Emerging research has unveiled the robustness of Vision Transformers (ViTs) over Convolutional Neural Networks (CNNs) (Shao et al., 2021; Paul & Chen, 2022), emphasizing the low transferability of adversarial instances between these architectures (Mahmood et al., 2021), despite some counter-points (Bai et al., 2021a). ViTs exhibit insensitivity to patch-based transformations that significantly distort original semantics, suggesting their reliance on robust yet non-indicative features Qin et al. (2022).

## 6 DISCUSSION

**Societal Impact** Although the name nullspace might imply a negative property, we notice the most practical implication of nullspace is to offer explanations to ViT's additional resilience toward minor noises in comparison to CNNs. Thus, we do not expect our work to introduce any negative societal impacts.

**Applications in Model Patenting** In addition to the applications we discussed, we consider another potential usage of our findings is to patent a ViT after a model is trained, as the nullspace will be unique property of any set of weights of certain ViT architectures. Different from the existing line of research in model watermarking (Adi et al., 2018; Darvish Rouhani et al., 2019; Le Merrer et al., 2020), the patenting through nullspace will not require any additional steps during training, although will face limited usage scenarios in comparison.

**Applications in Image Watermarking** Using the nullspace noise, it is possible to apply signatures onto input images without harming the output or operability of the networks. In the supplementary document, we present the cases where certain marks in form of nullspace noise can be superimposed on any desired input image. For the network, the output as well as the explanation generated with existing approaches remains unaltered. This gives user the choice to hide their data from misuse and/or re-distribution.

**Potential Limitation about the Nullspace Approximation** Different from the nullspace defined in linear algebra, the nullspace of the entire ViT can only be approximated because of the non-linearity in the network architecture. However, it is worthy mentioning that we can still calculate the exact nullspace of ViT if we only consider the patch embedding layer, through which, our results will qualitatively deliver the same message, but with quantitative differences.

## 7 CONCLUSION

In this work, we have explored the concept of nullspace in Vision Transformers (ViTs) to understand their robustness. Our findings demonstrate that a non-trivial nullspace indeed exists for Vision Transformers, a direct consequence of the patch embedding layer. This discovery implies that there are elements that, when added to an input, do not affect the output of the network, potentially offering an explanation for the robustness exhibited by ViTs. Moreover, we have extended the definition of nullspace, preserving a property that implies invariance of a mapping's output to input perturbations, and empirically identified a space that exhibits such property within the input space of the non-linear transformer encoder.

By linking the presence of nullspace with our extended definition to the general robustness of a network, we were able to devise a new approach to improve the robustness of ViTs. Our empirical results suggest that fine-tuning ViTs with the learnt nullspace noise can significantly enhance their robustness to a variety of robustness benchmarks. This method offers a new direction for data augmentation and model training, which could be beneficial for further research in the field.

Looking forward, there is more to explore in the vast landscape of Vision Transformers. Future research could focus on the development of efficient algorithms for learning nullspace and investigate its presence in other architectures and layers of deep neural networks. Our study offers a new perspective to the robustness of vision transformers. We believe these findings can assist in furthering the robustness of ViTs, potentially facilitating advancements in the development of more resilient machine learning models.

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

# A APPENDIX

## A.1 NULLSPACE: PRIMER

For a linear mapping such as $\beta$, where the domain is a vector space, nullspace is a subspace and satisfies all the required axioms:

- The zero element of $\mathbb{R}^{1 \times p}$, $\mathbf{0} \in \mathcal{N}$. This is true as $\mathbf{0}\beta = \mathbf{0}$.
- $\mathcal{N}$ is closed under vector addition. $\mathbf{u} + \mathbf{v} \in \mathcal{N}$ $\forall \mathbf{u}, \mathbf{v} \in \mathcal{N}$. This is true as $(\mathbf{u} + \mathbf{v})\beta = \mathbf{u}\beta + \mathbf{v}\beta = \mathbf{0}$.
- Closed under multiplication with scalar. $c\mathbf{u} \in \mathcal{N}$ $\forall\, c \in \mathbb{R}$ and $\mathbf{u} \in \mathcal{N}$. This is true as $(c\mathbf{u})\beta = c(\mathbf{u}\beta) = \mathbf{0}$.

A trivial nullspace, $\mathcal{N} = \{\mathbf{0}\}$, always exists for a linear mapping as described in equation 1. An alternate way to interpret nullspace is to view it as a set of solutions to the homogeneous system of linear equations as described by equation 1. This imples that $\mathbf{0}$ is always a solution to the said equation. As the number of solutions to a system of linear equations can vary, the nullspace for a mapping can be trivial or nontrivial.

## A.2 PROOF OF PROPOSITION 1

Let $d$ be the hidden dimension of the attention layer. $\mathbf{Q}_i, \mathbf{K}_i \in \mathbb{R}^{d \times d_k}$ where $d_k = d/h$. $\text{rank}(\mathbf{Q}_i \mathbf{K}_i^\top) \leq \text{rank}(\mathbf{K}_i^\top) \leq d_k$. Consider the sum of row spaces $S = \text{R}(\mathbf{Q}_1 \mathbf{K}_1^\top) + \text{R}(\mathbf{Q}_2 \mathbf{K}_2^\top) + \cdots + \text{R}(\mathbf{Q}_h \mathbf{K}_h^\top)$. $S$ is a subspace of $\mathbb{R}^d$. For $i = 1, \ldots, h$, choose a basis for $\text{R}(\mathbf{Q}_i \mathbf{K}_i^\top)$, denoted as $B_i = \{\mathbf{b}_1, \cdots, \mathbf{b}_{n_i}\}, |B_i| = n_i \leq d_k$. Without loss of generality, let $\mathbf{r}_{m,k} \in B_m$.

$S = \text{span}(\bigcup_{i=1}^{h} B_i)$, so

$$\dim(S) = \dim\left(\text{span}\left(\bigcup_{i=1}^{h} B_i\right)\right) = \dim\left(\text{span}\left(\left(\bigcup_{\substack{i=1 \\ i \neq m}}^{h} B_i\right) \cup (B_m \setminus \{\mathbf{r}_{m,k}\})\right)\right)$$

$$\leq \left|\left(\bigcup_{\substack{i=1 \\ i \neq m}}^{h} B_i\right) \cup (B_m \setminus \{\mathbf{r}_{m,k}\})\right| \leq (h-1)d_k + (d_k - 1) = d - 1$$

So, $\exists \mathbf{w} \in \mathbb{R}^d, \mathbf{w} \neq \mathbf{0}$ and $\mathbf{w} \in S^\perp$. This means for $i = 1, \ldots, h, \mathbf{w} \in \left(\text{R}\left(\mathbf{Q}_i \mathbf{K}_i^\top\right)\right)^\perp, \mathbf{w} \in \text{N}\left(\mathbf{Q}_i \mathbf{K}_i^\top\right)$. By condition 2, $\text{N}(\mathbf{V}_i) \supseteq \text{N}(\mathbf{Q}_i \mathbf{K}_i^\top)$, so $\mathbf{w} \in \text{N}\left(\mathbf{Q}_i \mathbf{K}_i^\top\right) \cap \text{N}(\mathbf{V}_i^\top)$.

Then, we can choose $\mathbf{W}$ wherein each row is a multiple of $\mathbf{w}$. We have $\mathbf{W}\mathbf{V}_i = \mathbf{0}$, and for any input to the encoder $\mathbf{X} \in \mathbb{R}^{n \times d}$,

$$\mathbf{W}\mathbf{Q}_i \mathbf{K}_i^\top \mathbf{X}^\top + \mathbf{X}\mathbf{Q}_i \mathbf{K}_i^\top \mathbf{W}^\top + \mathbf{W}\mathbf{Q}_i \mathbf{K}_i^\top \mathbf{W}^\top = \mathbf{0} \tag{12}$$

Consider the output of attention head

$$\text{head}_i(\mathbf{X} + \mathbf{W}) = \text{Softmax}\left(\frac{(\mathbf{X} + \mathbf{W})\,\mathbf{Q}_i \mathbf{K}_i^\top (\mathbf{X} + \mathbf{W})^\top}{\sqrt{d_k}}\right)(\mathbf{X} + \mathbf{W})\mathbf{V}_i$$

$$= \text{Softmax}\left(\frac{\mathbf{X}\mathbf{Q}_i \mathbf{K}_i^\top \mathbf{X}^\top + \mathbf{W}\mathbf{Q}_i \mathbf{K}_i^\top \mathbf{X}^\top + \mathbf{X}\mathbf{Q}_i \mathbf{K}_i^\top \mathbf{W}^\top + \mathbf{W}\mathbf{Q}_i \mathbf{K}_i^\top \mathbf{W}^\top}{\sqrt{d_k}}\right)\mathbf{X}\mathbf{V}_i$$

$$= \text{Softmax}\left(\frac{\mathbf{X}\mathbf{Q}_i \mathbf{K}_i^\top \mathbf{X}^\top}{\sqrt{d_k}}\right)\mathbf{X}\mathbf{V}_i = \text{head}_i(\mathbf{X})$$

Adding the noise $\mathbf{W}$ does not change the output of any attention head for arbitrary input $\mathbf{X}$, which completes our proof.

## A.3 Algorithm and implementation details

We present the algorithm of our data augmentation with nullspace noise in Algorithm 1.

---

**Algorithm 1:** Data augmentation with nullspace noise

---

1 **Input:** transformer model with patch embedding layer $f_e$, encoder $f_\phi$ and linear classifier $f_\psi$ parameterized by $e, \phi, \psi$ respectively; training data $\mathcal{T}$; batch size $B$; sampling limit $A$; noise nullity threshold $\epsilon$; noise learning rate $\eta_v$; model learning rate $\eta_f$; number of outer iterations $K$, noise training step $T$, model training step $S$

2 **for** $k = 0, \cdots, K-1$ **do**

3     Sample initial noise $\mathbf{v} \sim \mathrm{U}(-\mathrm{lim}, \mathrm{lim})$

4     **for** $t = 0, \cdots, T-1$ **do**

5         Sample a minibatch $(\mathbf{X}, \mathbf{y}) \sim \mathcal{T}$

6         Compute $\mathbf{U} \leftarrow f_e(\mathbf{X})$

7         Compute logits $\mathbf{Z} \leftarrow f_\psi(f_\phi^0(\mathbf{U})), \mathbf{Z}' \leftarrow f_\psi(f_\phi^0(\mathbf{U} + [\mathbf{v}]))$   `# "[v]" means broadcasting the noise v along the sample dimension`

8         Compute $\delta \leftarrow \frac{1}{B} \sum_{i=1}^{B} \|\mathrm{Softmax}(\mathbf{z}_i') - \mathrm{Softmax}(\mathbf{z}_i)\|^2$   `# z_i is sample logit`

9         **if** $\sigma < \epsilon$ **then**

10             **break**

11         **end**

12         Calculate $\ell \leftarrow \frac{1}{B} \sum_{i=1}^{B} \|\mathbf{z}_i' - \mathbf{z}_i\|^2$

13         Update $\mathbf{v} \leftarrow \mathbf{v} - \nabla_{\mathbf{v}} \ell$

14     **end**

15     **for** $s = 0, \cdots, S-1$ **do**

16         Sample a minibatch $(\mathbf{X}, \mathbf{y}) \sim \mathcal{T}$

17         Compute $\mathbf{U} \leftarrow f_e(\mathbf{X})$

18         Compute logits $\mathbf{Z} \leftarrow f_\psi(f_\phi^0(\mathbf{U})), \mathbf{Z}' \leftarrow f_\psi(f_\phi^0(\mathbf{U} + [\mathbf{v}]))$

19         Compute loss $\mathcal{L} \leftarrow \frac{1}{B} \sum_{i=1}^{B} (\ell(\mathbf{z}_i, y_i) + \ell(\mathbf{z}_i', y_i))$, where $\ell$ is the cross-entropy loss

20         Update model parameters $(\psi, \phi, e) \leftarrow (\psi, \phi, e) - \nabla_{(\psi, \phi, e)} \mathcal{L}$

21     **end**

22 **end**

23 **Output: model weight** $(\psi, \phi, e)$

---

**Hyperparameters** We fine-tuned the ViT model for $K = 20$ rounds in all settings. In each round, we initialized the noise with sampling limit $A = 3$, optimized it with learning rate $\eta_v = 0.1$ and set a threshold of $\epsilon = 0.03$. We empirically found that $T = 3000$ is enough to trigger early stopping so that the learned noise satisfies the $\epsilon$ threshold. We used $\eta_f = 10^{-5}$ to fine-tune the model for $S = 40$ iterations in each round. We set batch size $B = 128$, and slightly different from the vanilla SGD in Alg 1, we used the AdamW optimizer (Loshchilov & Hutter, 2019) and cosine learning rate scheduler with defualt hyperparameters for both the noise and the model training.

The original `ViT-B + DAT` model (Mao et al., 2022b) used the Exponential Moving Average (EMA) for evaluation[5], so we also used EMA to evaluate the performance of `ViT-B + DAT` fine-tuned with our method. For all the other settings, we used single model without ensemble for evaluation. We used $\epsilon = 1/255$ for the FGSM attack consistent with Mao et al. (2022b).

**Computation time** The experiments were conducted on a combination of A100, V100 GPUs and a 3090 GPU, depending on the availability. Although we only used about 10% of the ImageNet-1k (Deng et al., 2009) training data to fine-tune the model, the main computation time is on training the nullspace noise. One run of our experiment (20 rounds) takes the time roughly equivalent to 8 epochs of standard training on ImageNet-1k.

---

[5] https://github.com/alibaba/easyrobust

## A.4 NULLSPACE OF PRETRAINED NETWORKS

In Table 4 we list the nullspace dimension of the patch embedding layer for pretrained `ViT` models. For the cases where the embedding dimensions is greater than the patch size, the nullspace cannot exist (as per rank-nullity theorem), is 0.

Table 4: Empirically computed nullspace dimensions for the pre-trained `ViT` models.

| Model | Patch Size | Embedding Dimension | Nullspace Dimension |
|---|---|---|---|
| `ViT`-tiny | 16×16 | 192 | 576 |
| `ViT`-small | 32×32 | 384 | 2688 |
| `ViT`-small | 16×16 | 384 | 384 |
| `ViT`-base | 32×32 | 768 | 2304 |
| `ViT`-base | 16×16 | 768 | 2 |
| `ViT`-base | 8×8 | 768 | 0 |
| `ViT`-large | 32×32 | 1024 | 2048 |
| `ViT`-large | 16×16 | 1024 | 0 |

## A.5 PROPERTIES OF THE APPROXIMATE NULLSPACE

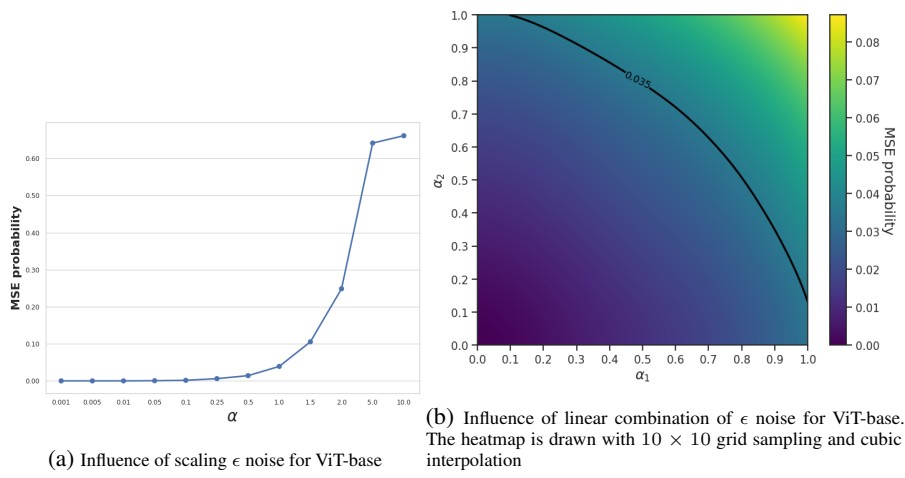

(a) Influence of scaling $\epsilon$ noise for ViT-base

(b) Influence of linear combination of $\epsilon$ noise for ViT-base. The heatmap is drawn with $10 \times 10$ grid sampling and cubic interpolation

Figure 4: Validation of the properties of the $\epsilon$ approximate nullspace

To explore the properties of the $\epsilon$-approximate nullspace beyond individually learned vectors, we studied their behaviors under scaling and linear combination. Using the same pretrained ViT-Base model, we repeatedly learn 98 $\epsilon$-approximate nullspace vectors with different random initializations. We set $\epsilon = 0.03$, but due to the early stopping in our algorithm, the MSE probability of the noise vectors are larger when evaluated on the whole validation set, ranging from 0.035 to 0.042. For the scaling experiment, we vary the scaling factor and keep track of the noise influence in terms of MSE probabilities, and take the average over all vectors. To study their linear combination, we take 10 different pairs of nullspace vectors $(v_1, v_2)$, take their linear combination $\alpha_1 v_1 + \alpha_2 v_2$, with $\alpha_1$ and $\alpha_2$ ranging between $[0, 1]$ with a grid size of $0.1$. We then evaluate the influence of the linearly combined noise at each point of the grid, averaged over all pairs of noise vectors.

The result in Fig. A.5 shows that the approximate nullspace has very similar property to vector space in terms of closure under addition and scalar multiplication. When the scaling factor $\alpha < 1$, we see a clear trend that the MSE probability of the scaled noise is less than $\alpha \epsilon$. In the linear combination case, the line $\alpha_1 + \alpha_2 = 1$ is well within the contour line of MSE probability being $0.035$, showing that the convex combination of a pair of $\epsilon$ noise vector is still $\epsilon$-approximate. These result shows interesting property of our approximate nullspace similar to the linear space.

### A.6 CHANGE TREND OF THE NOISE INFLUENCE WITH THE FINE-TUNING STEPS

Beside the trend of noise norm and performance metrics in Fig. 3, we also keep track of the influence of the learned noise in terms of MSE probability (2.3) at every 80 steps of the model fine-tuning. As shown in Table 5, the noise influence is always lower than $\epsilon = 0.03$, which means early stopping is triggered and the model enters the $\epsilon$ region.

Table 5: MSE probability of the noise at different fine-tuning steps.

| Fine-Tuning Step | 40 | 120 | 160 | 280 | 360 | 440 | 520 | 600 | 680 | 760 |
|---|---|---|---|---|---|---|---|---|---|---|
| MSE Probability | 0.028 | 0.027 | 0.026 | 0.029 | 0.028 | 0.029 | 0.027 | 0.025 | 0.028 | 0.026 |

### A.7 WATERMARKING IMAGES

Watermarking as image, usually used to convey ownership information or verify content of the data, has been studied extensively (Wolfgang & Delp, 1996; Potdar et al., 2005; Al-Haj, 2007; Bhat et al., 2010). A watermark can be either imperceptible or perceptible. and perceptible watermarking applies a noticeable marker to convey the protected nature of the data (Berghel, 1998). In this section, we explore to utilize nullspace noise to apply a perceptible watermark on the image which does not alter the test-time forward process.

Figure 5 shows an example watermarking approach using the nullspace noise. Here, we emboss the ICLR logo onto the natural images. The resulting modified image, attains the final predictions close to the original image. ($100\%$ match in the final output prediction and $10^{-4}$ difference in the predicted confidence value of the assigned class.)

**Method details:** With basis vectors of the nullspace, we can construct a watermark to be overlaid on the original image without affecting the output of the network. Given a source (user's image) and a target image (watermark), we simply need to estimate the scalar parameters corresponding to the basis vectors to satisfy $\sum_{i=0}^{i<m} \mathbf{e}_i \lambda_i = \mathbf{v}_\theta \approx \Delta \mathbf{x}_j$.

$\mathbf{e}_i$ are the basis vectors for the nullspace, $\lambda_i$ are their corresponding scalar co-efficients which are to be determined and $\Delta \mathbf{x}_j$ is the changed required to convert $j^{\text{th}}$ original image patch to $j^{\text{th}}$ watermark image patch. This can be achieved through a constrained optimisation of the following form:

$$\min \|\Delta \mathbf{x}_j - \sum_{i=0}^{i<m} \mathbf{e}_i \lambda_i\|_p \tag{13}$$

where, $\Delta x_j$ is the difference between the $j^{th}$ channel of a source and target image and $\lambda_i$ is the $i^{th}$ nullspace basis vector of the patch embedding layer with the corresponding variable scalar $e_i$. We use a least-square solver to solve for the solution (Available readily with packages such as Numpy).

### A.8 TARGETED NULLSPACE NOISE

Due to the dimension reduction effect of the patch embedding layer in most ViTs, we can transfer an image to be visually similar to another image by human perception, without changing the output of the original image perceived by the model. This differs from adversarial examples in the following aspects:

1. The working direction to construct an adversarial example is the other way around. If the transformed image is to be considered an adversarial example, then our source becomes the target for adversarial training and our target becomes the source.

2. Generating targeted nullspace noise requires no backpropagation through the network

3. Not only does the final prediction on the transformed image matches the source image, the saliency maps also match. This is displayed in Fig. 6

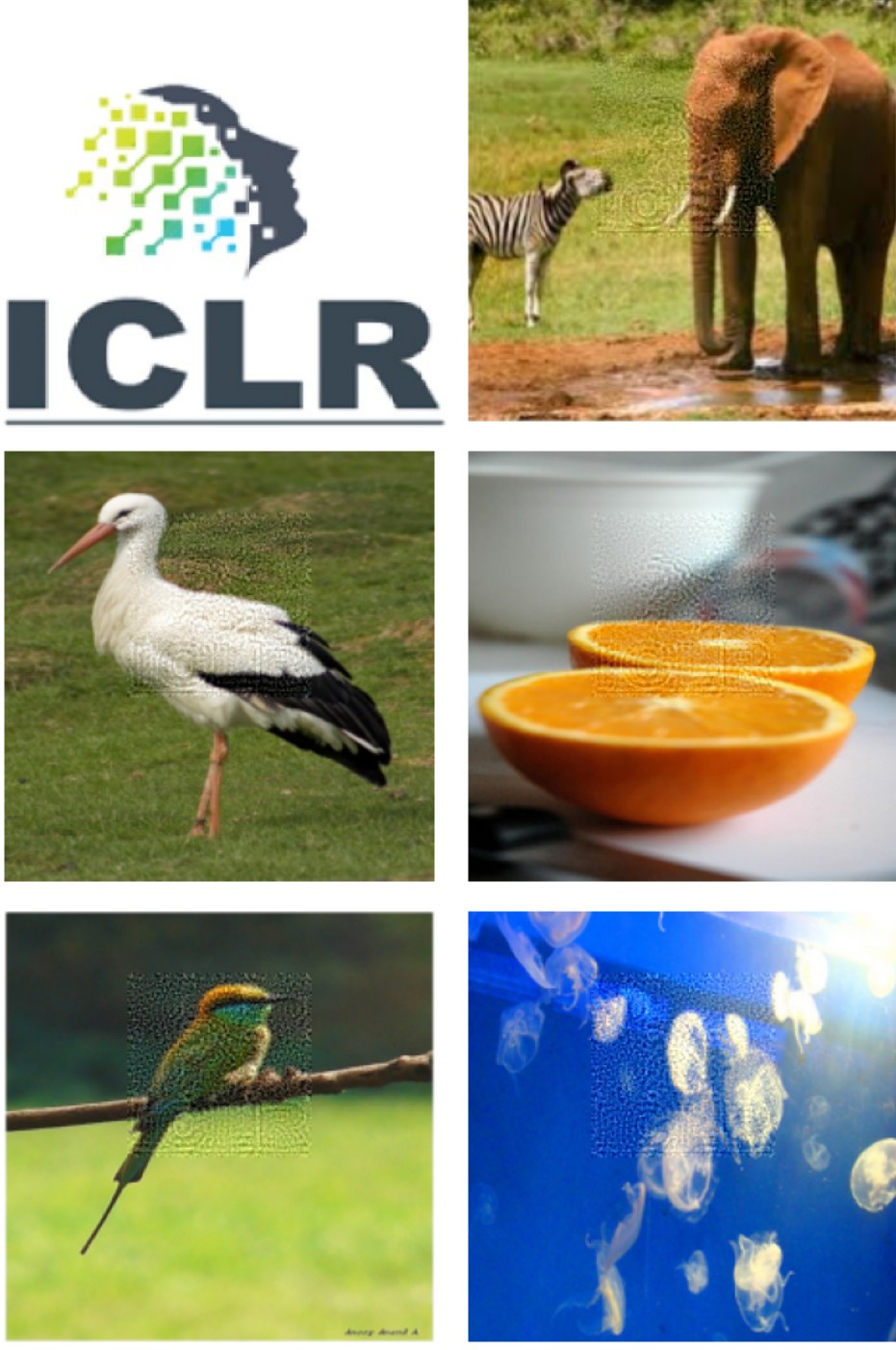

Figure 5: **Watermark superposition using the nullspace basis vectors.**(images changed to ICLR watermarks)

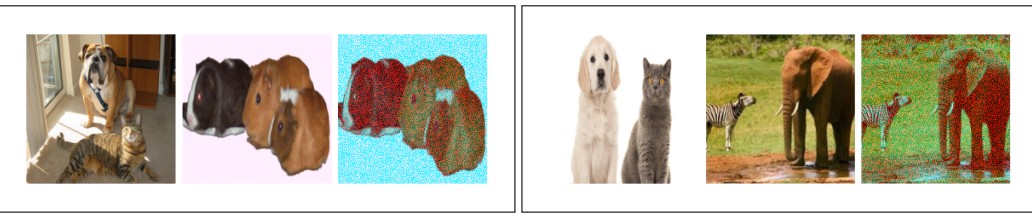

(a) Triplet of Source, target and transformed images

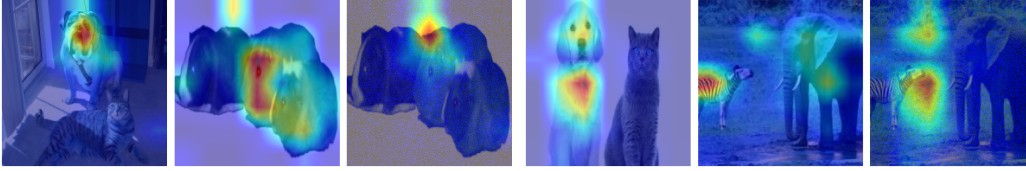

(b) Saliency maps for the corresponding images from the row above.

Figure 6: **Targeted nullspace noise.** Transformed images appear visually as target images but are interpreted as source images by the model. The equivalence between source and transformed images is not only in terms of the final predictions but also in the interpretability maps depicted in (b).

Though the transformation is not perfect, we can spot that the transformed images are visually similar to target images rather than source images. Even with this difference in the input space, transformed images and source images are classified into the same category with roughly the same confidence.

As recent studies have shown, fooling can also be extended to the interpretability methods (XAI) Dombrowski et al. (2019) partially due the limitations exposed by recent studies (Dombrowski et al., 2019; Ghorbani et al., 2019; Heo et al., 2019). However, in contrast to these works aiming to fool specific XAI method, our nullspace noise only depends on the model, not the XAI method.

In Fig. 6(b), we show the interpretability maps as generated by LRP (Chefer et al., 2021). From the figure, we can observe that the heatmaps generated by source and transformed images are identical whereas, the transformed image heatmaps substantially differ from target images'. Though only reported for LRP, we observed that a similar observation holds across different interpretability approaches. Here, we only presented the results on LRP, as in the context of ViTs, we found the heatmaps from other methods to be lacking (also pointed out by authors of LRP).

In Fig. 7 we show the saliency maps generated by different XAI methods. Even though the maps generated by methods other than LRP are poor (hard to interpret), we see that the source and transformed respond similarly to these methods.

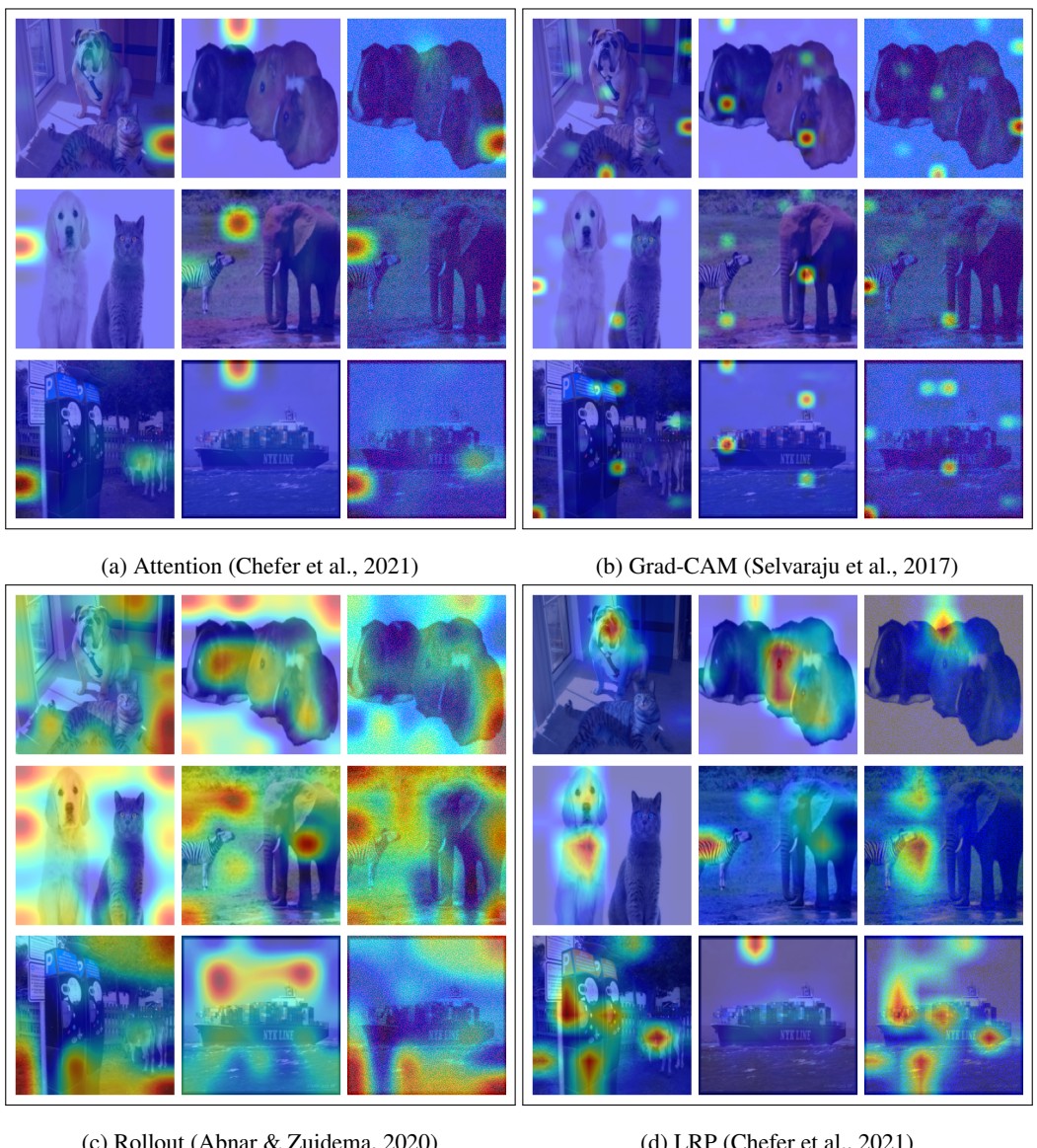

(a) Attention (Chefer et al., 2021)     (b) Grad-CAM (Selvaraju et al., 2017)

(c) Rollout (Abnar & Zuidema, 2020)     (d) LRP (Chefer et al., 2021)

Figure 7: **Interpretability maps generated via different methods for (source, target, transformed) images**

