# OpenReview forum: "Improving Robustness in Vision Transformers with Nullspace Noise Augmented Finetuning"
_ICLR.cc/2024/Conference — Submitted to ICLR 2024_

### Official Review · Reviewer_WMY4 · 2023-10-27

**Soundness:** 2 fair
**Presentation:** 1 poor
**Contribution:** 1 poor
**Rating:** 3
**Confidence:** 3

**Summary:**

In this paper, the authors show that transformers are invariant to certain input perturbation. This property, which they attribute to the nullspace property of the patch embedding layer, allows the authors to improve the robustness of vision transformers to adversarial input. In particular, the authors show that the output of the vision transformer may remain unchanged even under perturbations with large norms.

**Strengths:**

- The proposed approach is quite original. In particular, studying the nullspace of the patch embedding layer is novel and interesting.

**Weaknesses:**

- The mathematical contributions are minor and the paper contains some approximations (e.g., "closeness" instead of "closedness", last sentence of A1...).
- Despite being locally well written (i.e. taking sentences out of context, they are nicely written), the overall writing is confusing. It is difficult to keep what the authors are aiming to do in sight. For instance, Section 3 is cahotic and unfinished.
- As a consequence, it is difficult to evaluate the experimental contributions of this work.
- Some sentences convey an inapropriate tone, either overconfident or unscientific ("we offer a fresh narrative on...", "finding its nullspace is known art", "This observation leads to a provocative question", "Our work marks an important stride in the understanding of ...").

**Questions:**

I acknowledge the effort of the authors in this work, but I am afraid it suffers from serious drawbacks making it unpublishable in its current form.

**Major comments**
1. As underlined by the authors, the nullspace is a concept from linear algebra. When moving to the nonlinear world, its counterpart is the zero set. This is a common tool in convex optimization [1] which the authors may want to investigate. It seems however that rather than looking for a set of zeros, the authors are more interested in looking for invariance of the ViT wrt some additive subspaces (this is equation (5)). It may be interesting to change a bit the narrative to motivate it by invariant properties instead of null space properties?
2. Similarly, the authors state in the conclusion that "Our findings demonstrate that a non-trivial nullspace indeed exists" but I am not sure that the paper shows that. The authors show that a nullspace exists for the patch embedding layer, but then the full study resolves more around invariance properties of the ViT than its nullspace.
3. The authors write: "We ascertain that employing noise sampled from this approximate nullspace for data augmentation significantly enhances model robustness ...". This formulation suggests that the training of the ViT performance can improve when trained on data from its nullspace; however if the data shown to the network belong to its nullspace, how can it have an influence on its output?
4. I have an issue with loss (6). Unless I misunderstood, it is not lower bounded because no constraint is added. Thus, the minimum of (6) is reached by any vector of infinite norm. Why not add any constraint in the norm of $\tilde{v}$? I suspect this is why early stopping is needed (page 18)?
5. More generally, I find it interesting to look for vectors of relatively large norms, showing that very different inputs can provide the same output. However, a problem is that the resulting vector $u+\tilde{v}$, where $u$ is a natural image, is not a natural image. What are the potential applications in this context? Is it better for a model to be robust to nullspace variations than to common adversarial attacks?
6. Section 3 is very clearly unfinished, with unrelated sentences and equations. What is the narrative from (7) to (10)?
7. In section 3, the authors state that "we demonstrated that there may exist a non-isotropic space". Was it really shown that it is non isotropic?
8. Section 4.4 is very difficult to follow. What is the aim of the experiment presented there? For instance, the authors claim "the noise was always able to enter the $\epsilon$ region". But how can we be sure? Does any metric check that? This is not what Fig. 3 is showing, or I misunderstand?
9. The writing style is often inapropriate. I would suggest the authors adopt a more neutral tone. Here are few examples and why I think these are not appropriate:
     - "Our work marks an important stride in the understanding of [something]..." : this sounds presomptuous, readers will judge if this is "an important stride". Instead, the authors could say "Our work provides an explanation of [something] by ...". Similarly, "we offer a fresh narrative" sounds weird.
     - "This observation leads to a provocative question: Can the inherent properties of ViTs be harnessed to bolster their robustness?" First of all, I think the question is not provocative at all, it is typically the kind of questions that are asked in this conference. Secondly, the question could be reformulated in a less theatrical way. For instance: "This observation raises the following question: Can we leverage the inherent properties of ViTs to enhance their robustness?"

**Minor comments**

1. Some notations are unclear. For example, $f(u)[0]$ mixes mathematical and informatics notations. Why not use $f_0(u)$?
2. "This implies that 0 is always a solution to the said equation. As the number of solutions to a system of linear equations can vary, the nullspace for a mapping can be trivial, non-trivial, or does not exist." You just showed above that a trivial nullspace always exists for a linear mapping. How can the nullspace not exist?
3. "In terms of solving for $\mathcal{N}$?": why this question? I think that the footnote does not need to be introduced by a question.
4. For the remark after Proposition 1: there is no need to explain the proposition in the remark. I would suggest to only keep the informative example, and detail more on this example. Is this a condition that is met experimentally? Is this realistic?
5. Maybe the authors could change the NeurIPS watermark to an ICLR watermark?


**References**

[1] Bauschke, H. H., and P. L. Combettes. "Convex Analysis and Monotone Operator Theory in Hilbert Spaces, 2011.

---

> ### Author Response · Authors · 2023-11-19
> **Author Response  (1/3)**
>
> > **Q1: As underlined by the authors, the nullspace is a concept from linear algebra. When moving to the nonlinear world, its counterpart is the zero set. This is a common tool in convex optimization [1] which the authors may want to investigate. It seems however that rather than looking for a set of zeros, the authors are more interested in looking for invariance of the ViT wrt some additive subspaces (this is equation (5)). It may be interesting to change a bit the narrative to motivate it by invariant properties instead of null space properties?**
>
> Thank you for the insightful comment. Although the common definition of nullspace doesn’t apply to the nonlinear transformer encoder, we empirically find a set of vectors in the input space of the transformer encoder which preserves one property of nullspace that is important to robustness, i.e., invariance of the mapping’s output when added to any input, with good precision on the data distribution. Additionally, our experiment in Appendix A.5 suggests that the $\epsilon$-approximate version of such a set satisfies the properties of closure under addition and scalar multiplication in a considerable scope, similar to those of a vector space. Thus, we extend the definition of nullspace in linear algebra to “encoder-level nullspace” in Equation (5), to highlight the similarity between the explored set and nullspace in linear algebra.
>
> We’ve considered some well-established concepts in mathematics, such as “zero set” or “invariant set”, as alternative terminology, but they carry their own meanings and have different implications from our work. In Section 2.3 of our manuscript, we delineated the nuances of our extended definition with a focus on the invariance property.
>
> To address your concern, we have refined our writing in Section 2.3 and the conclusion to make the distinction more explicit.
>
> > **Q2.A: Similarly, the authors state in the conclusion that "Our findings demonstrate that a non-trivial nullspace indeed exists" but I am not sure that the paper shows that.**
>
> Thanks for this comment. The complete sentence in our paper was “Our findings demonstrate that a non-trivial nullspace indeed exists for Vision Transformers, a direct consequence of the patch embedding layer.” In section 2.2, we first showed that a non-trivial nullspace exists within the patch embedding layer. Then, because vectors in this set would not affect the entire model’s output when added to any input, this nullspace is a subset of the nullspace of the transformer model, according to our extended definition in Equation (5).
>
> > **Q2.B: The authors show that a nullspace exists for the patch embedding layer, but then the full study resolves more around invariance properties of the ViT than its nullspace.**
>
> Our discussion on the nullspace of the patch embedding layer, and on the nullspace of the nonlinear encoder according to our extended definition, are concrete definitions of invariance of the transformer model to input perturbations, in contrast to the broadly used “invariance”, which is often vaguely defined as model’s output does not change with certain (or even any) perturbations of input data.
>
> > **Q3: The authors write: "We ascertain that employing noise sampled from this approximate nullspace for data augmentation significantly enhances model robustness ...". This formulation suggests that the training of the ViT performance can improve when trained on data from its nullspace; however if the data shown to the network belong to its nullspace, how can it have an influence on its output?**
>
> Thanks for this question. We hope to clarify that our original claim quoted in the review is about “approximate nullspace”, for which our definition is the set of vectors which causes a mean $\ell_2$ distance of less than $\epsilon$ on the output logits when added to the input over the data distribution], as in Equation (7). It can have effects on the model by definition.  The goal is to reduce the impact of the identified approximate noise on the network and by doing this we make the model more robust.

---

> ### Author Response · Authors · 2023-11-19
> **Author Response (2/3)**
>
> > **Q4.A : I have an issue with loss (6). Unless I misunderstood, it is not lower bounded because no constraint is added. Thus, the minimum of (6) is reached by any vector of infinite norm.**
>
> Thank you for the observation. For \lambda < \infty, whether a vector of infinite norm is a minimizer of the entire loss function will also depend on the 1st term. For example, if we consider a simple case where both f in (6) are identity functions, then a vector of infinite norm is instead the maximizer.
>
> > **Q4.B Why not add any constraint in the norm of $\tilde{\mathbf{v}}$?**
>
> We agree this can potentially help prevent the discussions of extreme cases, however, in practice, we did not find it necessary. We noticed that the learning of $\tilde{\mathbf{v}}$ can always converge with the different values of $\lambda$ in our experiments.
> In addition, we had considered adding a bound constraint, but since we don’t have prior knowledge about how large the noise should at least be, we later decided to use a logarithm term to prevent a trivial solution.
>
> > **Q4.C  I suspect this is why early stopping is needed (page 18)?**
>
> In our work, early stopping was used as a heuristic to find vectors close to the boundary of the $\epsilon$-approximate nullspace. It is used in combination with Equation (6), which minimizes the influence of the noise with gradient descent until it reaches the $\epsilon$ region. Here, we no longer use the regularization term to prevent trivial solution.
>
> Additionally, the effectiveness of our data augmentation method does not rely on the early stopping heuristic. To validate this, we experimented on removing early stopping, and kept other settings the same as our experiment for row 2 of Table 1. We learn the nullspace noise with a fixed number of iterations T. We set T to be large enough so that the learned noise can enter the $\epsilon$ area every time after the fine-tuning starts. The below table shows the performance on various benchmarks, and compared it with the early stopping setting in row 2 of our Table 1.
>
> |  Early stop | clean | FGSM  | DamageNet |   A   | C (⭣) |  V2   |   R   | Sketch | Stylized |
> |:---------:|:-----:|:-----:|:---------:|:-----:|:-----:|:-----:|:-----:|:------:|:--------:|
> | Yes | 77.47 | 25.95 |   32.43    | 20.77  | 55.98 | 66.5   | 41.61 | 25.67  |  16.02   |
> |  No  | 77.01 | 28.84 |   30.26   | 19.68  | 57.68 | 66.77 | 44.21 | 29.40  |  17.31   |
>
> It can be seen that the performance of these two settings are comparable. We have added both the variants to the results table.
>
>
> > **Q5: More generally, I find it interesting to look for vectors of relatively large norms, showing that very different inputs can provide the same output. However, a problem is that the resulting vector $\mathbf{u} + \tilde{\mathbf{v}}$, where $\mathbf{u}$ is a natural image, is not a natural image. What are the potential applications in this context? Is it better for a model to be robust to nullspace variations than to common adversarial attacks?**
>
> Thank you for this inspiring question. When we find vectors of relatively large norms, the resulting vector $\mathbf{u} + \tilde{\mathbf{v}}$ can easily go beyond the pixel range of [0, 255]. However, there are still interesting applications. For example, we can consider these vectors a characteristic of the model being trained, and use it to identify the models, which is related to the application of model patenting. Another interesting aspect that can arise is that of gaining customer trust with regards to privacy. If a server-side ML model can support “unnatural” inputs, a user can effectively hide(transform) their data while preserving the intended inference output. Making it challenging to recover original “natural” images if/when leaked or stolen.

---

> ### Author Response · Authors · 2023-11-19
> **Author Response (3/3)**
>
> > **Q6: Section 3 is very clearly unfinished, with unrelated sentences and equations. What is the narrative from (7) to (10)?**
>
> We did complete the writing of Section 3, and we are sorry for any confusion brought to readers. We have added a few transition sentences and explanations in our paper, to improve the coherence of discussion.
> In addition to the text edited in the main paper, here are some additional explanations of Section 3.
>
> Equation (7) defines the space of additive perturbations that have minute influence to the model’s output. We use $\epsilon$ as the upper bound of the influence measured by MSE of the output logits, so that the space is specified with a given $\epsilon$ and a given model;
>
> Equation (8) defines the problem of finding the largest vector in this space as a reflection of its size;
>
> Equation (9) specifies the loss function used to learn the nullspace noise, which is used in combination with the early stopping method to heuristically solve the problem in Equation (8);
>
> Equation (10) specifies the loss function used in our data augmentation method, in which we used the noise learned by Equation (9) and early stopping to conduct data augmentation.
>
>
>
> > **Q7: In section 3, the authors state that "we demonstrated that there may exist a non-isotropic space". Was it really shown that it is non isotropic?**
>
> Yes, in Section 2.3, we measured the influence of the learned vectors in terms of MSE probability and percentage of matching predictions. We also experimented with resetting their directions by randomly shuffling their elements. The real and dashed lines in Fig. 2 shows that while the learned vectors are benign to the model, their shuffled version significantly degrades the model’s predictions at different values of $\lambda$. This shows that the noise vectors produce far less influence on the model’s output than they would if they were in a random direction with the same norm, which indicates that the space is non-isotropic.
>
> > **Q8: Section 4.4 is very difficult to follow. What is the aim of the experiment presented there? For instance, the authors claim "the noise was always able to enter the  region". But how can we be sure? Does any metric check that? This is not what Fig. 3 is showing, or I misunderstand?**
>
>   > **What is the aim of the experiment presented there?**
>
> The aim of this experiment is to use a data augmentation approach to enlarge the $\epsilon$-approximate nullspace of the model and see its effect on the model’s robustness. This is to explore the relationship between the nullspace and robustness.
>
>
>   > **For instance, the authors claim "the noise was always able to enter the  region". But how can we be sure?**
>
> The claim "the noise was always able to enter the region" was indeed not reflected in Fig. 3, but we carefully verified from the output of our experiment scripts that early stopping was triggered, which means the noise entered the $\epsilon$-approximate region. We have added the noise influence evaluation in Appendix A.6.
>
>   > **Does any metric check that?**
>
> The early stopping for learning noise our method (Section 3) is based on the condition that the noise has an influence less than $\epsilon$ in terms of MSE probability. We carefully verified from our experiment logs that early stopping was always triggered after the fine-tuning started, which means the noise entered the $\epsilon$-approximate region.
>
>   > **This is not what Fig. 3 is showing, or I misunderstand?**
>
> We hope to clarify the misunderstanding. Fig. 3 did not show this piece of information, but only showed the change trend of the noise norm and several performance metrics with the fine-tuning steps. We have updated the draft in Section 4.4 to improve clarity.
>
> > **Q9: The writing style is often inapropriate. I would suggest the authors adopt a more neutral tone. Here are few examples and why I think these are not appropriate:**
>
> Thank you for the suggestions. We have examined the writing style of our whole paper, and modified multiple places for a more neutral and humble tone, including the examples you provided.
>
> > **Minor comments**
>
> Thank you for providing a thorough review of our work. We acknowledge the minor comments and have subsequently updated our paper to reflect the same. We will be uploading the latest draft of our work in a short while.

---

> > ### Comment · Reviewer_WMY4 · 2023-11-21
> >
> > I thank the authors for their thorough response and effort in revising the paper. Overall, I find that the readability of the paper is improved. Please find below my remaining questions.
> >
> > **Q. 4a** The argument of the authors holds for $f_{\psi}\circ f_{\phi} = \operatorname{Id}$. Yet, the authors are precisely looking for a vector $v$ such that $f_{\psi}(f_{\phi} (u+v))-f_{\psi}(f_{\phi} (u))= 0$, which suggests that $f_{\psi}\circ f_{\phi} \neq \operatorname{Id}$. Assume that such $v^*\neq 0$ exists; if $\forall \lambda>0$, $f_{\psi}(f_{\phi} (u+\lambda v^*))-f_{\psi}(f_{\phi} (u))= 0$, then $\mathcal{L}(\lambda v^*)\underset{\lambda\to\infty}{\to}-\infty$. Moreover, if $f_{\psi}\circ f_{\phi}$ contains some thresholding operations, ensuring that $f_{\psi}\circ f_{\phi}$ is bounded, then the infimum of $\mathcal{L}$ is $-\infty$. This is problematic because nothing tells a priori whether the loss function aims at ensuring $f_{\psi}(f_{\phi} (u+v))-f_{\psi}(f_{\phi} (u))= 0$ or $||v||\to\infty$, and it may return a vector of large (infinite) norm that does not at all satisfy $f_{\psi}(f_{\phi} (u+v))-f_{\psi}(f_{\phi} (u))= 0$.
> >
> > **Q. 6** Section 3 is a little bit improved but remains unclear. For example, equation (9) is between (but not included in) two independent sentences. Equation (10) is starting a sentence. I think this section would be significantly clearer if (9) and (10) were written in a whole bilevel problem.
> >
> > **Training loss** The training loss is not explicitely mentionned in the experimental part. "We use the $\epsilon$-approximate nullspace noise as data augmentation to fine-tune the model." What loss is used to do that? More generally, it is difficult to relate the experimental part to the equations presented in the paper. This is fairly problematic: if the authors use equation (6), then **Q. 4a** above questions the full approach. However, if eq. (7) and (8) are used, then this is fine.

---

> ### Author Response · Authors · 2023-11-21
> **Author Response to Follow-up Questions**
>
> We sincerely appreciate your continued effort in reviewing our submission, and your insightful follow-up questions. Below are our responses:
>
> > Q. 4a The argument of the authors holds for $f_{\psi} \circ f_{\phi} = \text{Id}$. Yet, the authors are precisely looking for a vector $v$ such that $f_{\psi}(f_{\phi}(u + v)) - f_{\psi}(f_{\phi}(u)) = 0$, which suggests that $f_{\psi} \circ f_{\phi} \neq \text{Id}$.
>
> We agree that this particular example does not allow non-zero vectors to satisfy $f_{\psi}(f_{\phi}(u + v)) - f_{\psi}(f_{\phi}(u)) = 0$. However, we used the identity function as a conceptual example to show that whether Equation (6) goes to negative infinity as the noise vector scales depends on whether the first term grows more rapidly than the second term.
>
> > Assume that such $v^\star \neq 0$ exists; if $\forall \lambda > 0, f_{\psi}(f_{\phi}(u + \lambda v^\star)) - f_{\psi}(f_{\phi}(u)) = 0$, then $L(\lambda v^\star*) \rightarrow -\infty$ as $\lambda \rightarrow \infty$.
>
> Because the pre-Softmax logits are unbounded, increasing $\lambda$ would make the first term very large during the optimization process, but the logarithm term would not change much. In our experiments with Equation (6), we found that the noise vectors soon converged to a finite value under different regularization coefficients.
>
>
> > Moreover, if $f_{\psi} \circ f_{\phi}$ contains some thresholding operations, ensuring that $f_{\psi} \circ f_{\phi}$ is bounded, then the infimum of $L$ is $-\infty$.
>
> Our specific choice of $f_{\psi} \circ f_{\phi}$ is the transformer model, which takes patch embeddings as input and outputs the pre-Softmax logits. The pre-Softmax logits of a neural network can be unbounded by design. Thus as $\lambda$ goes to infinity, the first term in Equation (6) can be unbounded and grows more rapidly than the logarithm term, this may explain why our experiment with Equation (6) converges well and get the noise vectors with the desired invariance property.
>
> > This is problematic because nothing tells a priori whether the loss function aims at ensuring $f_{\psi}(f_{\phi}(u + v)) - f_{\psi}(f_{\phi}(u)) = 0$ or $\|v\| \rightarrow \infty$, and it may return a vector of large (infinite) norm that does not all satisfy $f_{\psi}(f_{\phi}(u + v)) - f_{\psi}(f_{\phi}(u)) = 0$.
>
> We agree that our formulation in Equation (6) does not explicitly prevent a vector with an infinitely large norm that does not meet the invariance condition. We agree that “it may return a vector of” that property, however, we hope to remind the reviewer that our empirical findings, depicted in Fig. 2, indicate that the loss term combined with the negative logarithmic term successfully yields non-trivial noise vectors that minimally impact the model output, verified by the MSE of probability and prediction consistency metrics. This approach aligns with the goals of our exploratory study to ascertain the feasibility of learning such noise vectors.
>
>
> We appreciate the reviewer's discussion on the statistical properties of loss. We agree that loss function design can benefit more from an explicit constraint, and we have added a remark at the top of this page to note future readers. Our current formulation leads to an empirically working method, and it meets the needs of our preliminary experiment. We hope the empirical contribution of our work will not be ignored because of this.
>
> > Q. 6 Section 3 is a little bit improved but remains unclear. For example, equation (9) is between (but not included in) two independent sentences. Equation (10) is starting a sentence. I think this section would be significantly clearer if (9) and (10) were written in a whole bilevel problem.
>
> Thank you for your insightful suggestion. We have modified our formulation and writing to improve the clarity of our method part.
>
> > Training loss The training loss is not explicitely mentionned in the experimental part. "We use the $\epsilon$-approximate nullspace noise as data augmentation to fine-tune the model." What loss is used to do that? More generally, it is difficult to relate the experimental part to the equations presented in the paper. This is fairly problematic: if the authors use equation (6), then Q. 4a above questions the full approach. However, if eq. (7) and (8) are used, then this is fine.
>
> Thanks for your instructive comment. We have added a reference to the equation in the experiment part in Section 4. We used Equation (7) and (8) for this experiment, which are Equation (7) and (9) in the updated draft. Equation (6) was only used in the exploratory experiment in Section 2.3.

---

### Official Review · Reviewer_LAzm · 2023-10-30

**Soundness:** 4 excellent
**Presentation:** 3 good
**Contribution:** 3 good
**Rating:** 8
**Confidence:** 2

**Summary:**

The paper presents a novel concept of "nullspace" in the context of Vision Transformers (ViTs) and explores its potential applications in improving ViT robustness. The paper shows that ViTs possess a non-trivial nullspace and demonstrates how learning "nullspace noise" can enhance robustness. Experimental results indicate that fine-tuning ViTs with learned nullspace noise leads to significant improvements in robustness against various benchmarks.

**Strengths:**

- The paper introduces an innovative concept, "nullspace," in ViTs, expanding our understanding of these models.
- The methodology used to identify nullspace and the experiments conducted to demonstrate its impact are comprehensive.
- The paper highlights the practical applications of nullspace, such as model patenting and image watermarking.

**Weaknesses:**

- The paper's discussion on societal impact and practical applications could benefit from further elaboration and real-world examples to strengthen the argument.
- While the nullspace concept is intriguing, the limitations of its applicability due to non-linearity in ViTs should be addressed more explicitly.

**Questions:**

- Can the authors provide specific real-world examples and use cases where the concept of nullspace and nullspace noise can be applied in practice?
- The paper mentions the non-linearity in ViTs as a limitation to the exact calculation of nullspace. Could the authors discuss potential methods or approaches to address this limitation or provide more insights into the nature of this non-linearity?

---

> ### Author Response · Authors · 2023-11-18
> **Author Response**
>
> **We sincerely appreciate your kind comments and positive assessment. In particular, we are grateful for your recognition of the novelty of our research topic, the soundness of the proposed method, and your interest in the application scenarios of our work. Below are our responses to your questions.**
>
>
> > **W1 & Q1: Can the authors provide specific real-world examples and use cases where the concept of nullspace and nullspace noise can be applied in practice?**
>
> Below we provide some examples about the applications of nullspace in ViTs in various societal and technological contexts.
>
> **Enhancing ViT's Resilience to Minor Noises:** This has practical implications in real-world scenarios where visual data might be subject to various types of noise, such as in low-light conditions or with low-quality sensors. Understanding and leveraging nullspace properties can enhance the robustness of ViTs in these noisy environments.
>
> **Model Patenting:** The unique nullspace property of a ViT, determined by its specific set of weights, can be used for patenting purposes. It allows the identification of a specific ViT model based on its nullspace characteristics, which can be particularly useful in legal scenarios involving intellectual property rights and model authentication.
>
> **Image Watermarking:** Nullspace noise can be used for watermarking images in a way that does not alter the output or operability of ViTs. By superimposing marks in the form of nullspace noise onto input images, users can protect their data from misuse or unauthorized redistribution. This application is particularly relevant in fields like digital media, where copyright protection is critical, and in sensitive applications like personal data security.
>
> > **W2 & Q2: The paper mentions the non-linearity in ViTs as a limitation to the exact calculation of nullspace. Could the authors discuss potential methods or approaches to address this limitation or provide more insights into the nature of this non-linearity?**
>
> There is inherent non-linearity in ViTs due to elements like activation functions and self-attention mechanisms, posing challenges to the exact analysis on the encoder-level nullspace.  In our work, we address this through approximate nullspace computation, employing deep-learning based optimization to identify minimally impactful vectors on the model’s output. This approach offers a meaningful approximation within the ViT framework, providing an understanding of the robustness of transformer models. Further research could delve into a more detailed analysis of ViTs, to study the nullspace within each layer and their cascading effect, or employ advanced optimization techniques to refine this approximation in non-linear systems.

---

### Official Review · Reviewer_43dA · 2023-11-01

**Soundness:** 2 fair
**Presentation:** 3 good
**Contribution:** 3 good
**Rating:** 5
**Confidence:** 4

**Summary:**

The paper connects the concept of nullspace with the robustness of vision transformers. In this paper the authors aims to identify the approximate nullspace for the ViTs at encoder level and then further proposes a fine-tuning method in which the ViTs would be trained with the synthetic patch embedding by adding the original patch embedding with the noise sampled from the nullspace. The experimental results show that ViTs trained by the proposed method can experience the better performance under adversarial attacks and distribution shift.

**Strengths:**

1. The papaer is well organized and written. The idea is easy to follow.

2. The idea that addresses the adversarial robustness of ViTs from perspective of nullspace is novel and interesting.

3. The paper provides sound theoretical foundation for the proposed method.

4. The proposed method achieves good performance under adversarial attacks and distribution shift.

**Weaknesses:**

1.For adversarial attacks, the authors only adopt FGSM and DamageNet. However, FGSM is a light-weight adversarial attack. Stronger attacks like PGD should also be considered. The performance of the model under stronger adversarial attack is a better measurement of the robustness of the model.

2.The baseline methods listed in the paper are insufficient. The author should also incoporate some other important baselines[1-5].

[1]Li, Yanxi, and Chang Xu. "Trade-Off Between Robustness and Accuracy of Vision Transformers." Proceedings of the IEEE/CVF Conference on Computer Vision and Pattern Recognition. 2023.

[2]Zhou, Daquan, et al. "Understanding the robustness in vision transformers." International Conference on Machine Learning. PMLR, 2022.

[3]Paul, Sayak, and Pin-Yu Chen. "Vision transformers are robust learners." Proceedings of the AAAI conference on Artificial Intelligence. Vol. 36. No. 2. 2022.

[4]Wu, Boxi, et al. "Towards efficient adversarial training on vision transformers." European Conference on Computer Vision. Cham: Springer Nature Switzerland, 2022.

[5]Chefer, Hila, Idan Schwartz, and Lior Wolf. "Optimizing relevance maps of vision transformers improves robustness." Advances in Neural Information Processing Systems 35 (2022): 33618-33632.

3.The paper claims that the proposed method can significantly improve robustness against adversarial and out-ofdistribution scenarios. Though the improvement on distribution shift is prominent, the performance under the adversarial attack is not promising.

**Questions:**

1.The proposed method seems independent on the model architecture. Would CNNs also experience improved robustness for the proposed method?

2.The patch-level attack [1.2] proves to be effective in attacking the ViT models. Is the proposed method also effective in defending these attacks?

[1] Gu, Jindong, Volker Tresp, and Yao Qin. "Are vision transformers robust to patch perturbations?." European Conference on Computer Vision. Cham: Springer Nature Switzerland, 2022.

[2] Fu, Yonggan, et al. "Patch-Fool: Are Vision Transformers Always Robust Against Adversarial Perturbations?." International Conference on Learning Representations. 2021.

---

> ### Author Response · Authors · 2023-11-18
> **Author Response**
>
> **We sincerely appreciate your kind comments and insightful suggestions. In particular, we are grateful for your recognition of our idea, theoretical derivation, and the empirical performance. Below are our responses to one of your comments**
>
> > **W1: For adversarial attacks, the authors only adopt FGSM and DamageNet. However, FGSM is a light-weight adversarial attack. Stronger attacks like PGD should also be considered. The performance of the model under stronger adversarial attack is a better measurement of the robustness of the model.**
>
> Thank you for the suggestion. Acknowledging the feedback, we have now incorporated CW, MIM and PatchFool. The results are shared at the end.
>
> > **W2: The baseline methods listed in the paper are insufficient. The author should also incorporate some other important baselines[1-5].**
>
> We appreciate the recommendations of the reviewer. We have added these baseline results into our Table 1.
> We appreciate your other comments about supplementary experiments. However, it would take us some time to address them. We have started doing these experiments, and will let you know as soon as we get some results.
>
> > **W3: The paper claims that the proposed method can significantly improve robustness against adversarial and out-of distribution scenarios. Though the improvement on distribution shift is prominent, the performance under the adversarial attack is not promising.**
>
> Our method trains the model to be tolerant to approximate nullspace vectors, which are non-trivial perturbations having minute influence on the model output learned from the model itself. We share below results on more adversarial attacks.
>
> > **Q1: The proposed method seems independent on the model architecture. Would CNNs also experience improved robustness for the proposed method?**
>
> Acknowledging the inquiry, we are currently running the experiments with CNN based finetuning and will provide an update in the upcoming days.
>
> > **Q2: The patch-level attack [1.2] proves to be effective in attacking the ViT models. Is the proposed method also effective in defending these attacks?**
>
> We are grateful for the question. We share evaluation on PatchFool below. From the results it is evident that our method helps against patch based attacks as well.
>
> |               |  CW  |  MIM(lower better)  | PatchFool |
> |---------------|:----:|:-----:|:---------:|
> | ViT-Base      | 0.56 | 81.72 |   15.92   |
> | ViT-Base + NS | 2.36 | 74.72 |   23.52   |
> | DAT           | 0.76 | 70.28 |   22.64   |
> | DAT + NS      | 0.88 | 71.18 |   24.14   |

---

> > ### Author Response · Authors · 2023-11-20
> > **Author Response with New Experimental Results**
> >
> > > **Q1: The proposed method seems independent on the model architecture. Would CNNs also experience improved robustness for the proposed method?**
> >
> > Thank you for your insightful question regarding the applicability of our nullspace data augmentation method to CNN architectures. We appreciate the opportunity to clarify this aspect.
> >
> > Our initial experiments with CNN models, specifically ResNet-18 and ResNet-50, have yielded mixed results. We observed a decrease in performance for ResNet-18 and a marginal improvement for ResNet-50 when applying our nullspace augmentation method. This variance in effectiveness is likely due to the architectural differences between CNNs and ViTs.
> >
> > |  Model   | nullspace | clean | FGSM  | DamageNet |   A   | C (↓) |  V2   |   R   | Sketch | Stylized |  Avg  |
> > |:--------:|:---------:|:-----:|:-----:|:---------:|:-----:|:-----:|:-----:|:-----:|:------:|:--------:|:-----:|
> > | ResNet18 |    No     | 71.55 | 6.37  |   6.40    | 2.45  | 77.42 | 59.45 | 34.64 | 23.65  |   7.72   | 25.75 |
> > | ResNet18 |    Yes    | 67.38 | 3.06  |   6.70    | 1.69  | 78.03 | 54.15 | 31.20 | 16.49  |   8.48   | 23.46 |
> > | ResNet50 |    No     | 80.14 | 23.92 |   17.49   | 11.92 | 63.56 | 68.66 | 39.71 | 28.77  |   8.23   | 34.82 |
> > | ResNet50 |    Yes    | 78.16 | 28.00 |   19.61   | 8.88  | 62.82 | 66.35 | 40.33 | 30.37  |  10.07   | 35.44 |
> >
> > Our method, particularly the learning of nullspace vectors, is tailored for the input space of the non-linear encoder of transformers. In contrast, CNN models lack distinct counterparts to the patch embedding and encoder components found in ViTs. In our experiment with CNNs, we adapted our method to learn nullspace vectors at the input space of the CNN. While our method has shown effectiveness for transformer models, its direct application to CNNs is not as straightforward and requires further investigation and adaptation to account for the architectural differences.

---

> > > ### Author Response · Authors · 2023-11-22
> > >
> > > Dear reviewer 43dA,
> > >
> > > As the discussion period draws to a close, we are particularly keen to hear your thoughts and feedback on our recent rebuttal response. Your expertise and insights are crucial for us to refine and improve our work. If you require any further information or clarifications to facilitate your review, please let us know.
> > >
> > > Thank you for your valuable time and attention to our work. We look forward to hearing from you soon.
> > >
> > > Sincerely,
> > >
> > > Authors

---

### Official Review · Reviewer_hjZY · 2023-11-11

**Soundness:** 3 good
**Presentation:** 3 good
**Contribution:** 2 fair
**Rating:** 5
**Confidence:** 4

**Summary:**

This paper propose to investigate adverial robustness of vision transformer by analyzing nullspace. Nullspace is found by optimization and further leveraged as data augmentation for training. The method is validated on various dataset  under both white and black box attacks.

**Strengths:**

1. The idea of using nullspace is quite interesting.
2. The paper is clearly organized and well written.
3. The derivation of existence of encoder-level nullspace is nice.

**Weaknesses:**

1. Why using nullspace as augmentation method could boost adversarial robustnes is unclear. As paper said, "We hypothesize that the model’s tolerance to approximate nullspace noise is indicative of its robustness under a variety of distribution shifts."  It's quite confusing, as the method is trying to find vectors that does not change model output, then regularize model to show no output change when adding such vector to input. Meanwhile, adversarial training finds perturbs that changes ouput most significantly and optmize the model to tolerate such perturb. The idea is somhow opppsite. Please justify why your method could work.
2. The approach could direclty applied on CNNs, though it is derived on ViT. Please check the performance on CNNs.
3. The experiments are weak. For white-box robustness, please use Auto-attack, CW attack, etc to evaluate. For black-box attack, please evaluate transferability from other models, using more well-known approaches such as MIM series attacks.

**Questions:**

Please refer to questions in weakness.

---

> ### Author Response · Authors · 2023-11-18
> **Author Response**
>
> **We sincerely appreciate your kind comments and insightful suggestions. In particular, we are grateful for your positive assessments of our idea, writing, and derivation. Below are our responses your concerns.**
>
> > **W1: Why using nullspace as augmentation method could boost adversarial robustnes is unclear. As paper said, "We hypothesize that the model’s tolerance to approximate nullspace noise is indicative of its robustness under a variety of distribution shifts." It's quite confusing, as the method is trying to find vectors that does not change model output, then regularize model to show no output change when adding such vector to input. Meanwhile, adversarial training finds perturbs that changes ouput most significantly and optmize the model to tolerate such perturb. The idea is somhow opppsite. Please justify why your method could work.**
>
> In our approach, we identify approximate nullspace vectors, which does not alter model outputs in theory but still introduces a small error, and then we train our models to make the output the same. This is based on the assumption that a robust model should be insensitive to perturbations that do not modify the semantic content of an image. On the other hand, adversarial training identifies the perturbation that changes the output the most under a norm constraint, and then trains the model to have the same output. This is also based on the assumption that a robust model should be insensitive to small perturbations. Both methods follow the logic of making the model tolerant to additive perturbations that should not alter its predictions, so our method shows the effect of improving the adversarial robustness similar to adversarial training.
>
> > **W2: The approach could direclty applied on CNNs, though it is derived on ViT. Please check the performance on CNNs.**
>
> Yes, this approach can be directly applied to CNNs as well. We are currently performing the training and will share the results in the next few days.
>
> > **W3: The experiments are weak. For white-box robustness, please use Auto-attack, CW attack, etc to evaluate. For black-box attack, please evaluate transferability from other models, using more well-known approaches such as MIM series attacks.**
>
> Thank you for the suggestions. We share below the results on CW, PatchFool[1] and MIM (using deit-tiny) for the ViT-Base. We have also added these results and experimental details to the adversarial robustness experiments of our paper which we will be uploading in a shortwhile.
>
> |               |  CW  |  MIM(fooling rate, lower better)  | PatchFool[1] |
> |---------------|:----:|:-----:|:---------:|
> | ViT-Base      | 0.56 | 81.72 |   15.92   |
> | ViT-Base + NS | 2.36 | 74.72 |   23.52   |
> | DAT           | 0.76 | 70.28 |   22.64   |
> | DAT + NS      | 0.88 | 71.18 |   24.14   |
>
> **References:**
>
> [1] Fu, Y., Zhang, S., Wu, S., Wan, C., & Lin, Y. (2021, October). Patch-Fool: Are Vision Transformers Always Robust Against Adversarial Perturbations?. In International Conference on Learning Representations.

---

> > ### Author Response · Authors · 2023-11-20
> > **Author Response with New Experimental Results**
> >
> > > **W2: The approach could direclty applied on CNNs, though it is derived on ViT. Please check the performance on CNNs.**
> >
> > Thank you for your insightful question regarding the applicability of our nullspace data augmentation method to CNN architectures. Below are our responses.
> >
> > Our initial experiments with CNN models, specifically ResNet-18 and ResNet-50, have yielded mixed results. We observed a decrease in performance for ResNet-18 and a marginal improvement for ResNet-50 when applying our nullspace augmentation method. This variance in effectiveness is likely due to the architectural differences between CNNs and ViTs.
> >
> > |  Model   | nullspace | clean | FGSM  | DamageNet |   A   | C (↓) |  V2   |   R   | Sketch | Stylized |  Avg  |
> > |:--------:|:---------:|:-----:|:-----:|:---------:|:-----:|:-----:|:-----:|:-----:|:------:|:--------:|:-----:|
> > | ResNet18 |    No     | 71.55 | 6.37  |   6.40    | 2.45  | 77.42 | 59.45 | 34.64 | 23.65  |   7.72   | 25.75 |
> > | ResNet18 |    Yes    | 67.38 | 3.06  |   6.70    | 1.69  | 78.03 | 54.15 | 31.20 | 16.49  |   8.48   | 23.46 |
> > | ResNet50 |    No     | 80.14 | 23.92 |   17.49   | 11.92 | 63.56 | 68.66 | 39.71 | 28.77  |   8.23   | 34.82 |
> > | ResNet50 |    Yes    | 78.16 | 28.00 |   19.61   | 8.88  | 62.82 | 66.35 | 40.33 | 30.37  |  10.07   | 35.44 |
> >
> > Our method, particularly the learning of nullspace vectors, is tailored for the input space of the non-linear encoder of transformers. In contrast, CNN models lack distinct counterparts to the patch embedding and encoder components found in ViTs. In our experiment with CNNs, we adapted our method to learn nullspace vectors at the input space of the CNN.

---

> ### Author Response · Authors · 2023-11-22
>
> Dear reviewer hjZY,
>
> As the discussion period draws to a close, we are particularly keen to hear your thoughts and feedback on our recent rebuttal response. Your expertise and insights are crucial for us to refine and improve our work. If you require any further information or clarifications to facilitate your review, please let us know.
>
> Thank you for your valuable time and attention to our work. We look forward to hearing from you soon.
>
> Sincerely,
>
> Authors

---

### Author Response · Authors · 2023-11-21
**Additional Remark to Remind Future Readers**

This paper has been revised following peer review. The authors acknowledge the insightful comments from all reviewers, which have been instrumental in refining both the empirical and theoretical aspects of the work.

The results of our preliminary experiments, notably illustrated in Fig. 2, demonstrate the effectiveness of our proposed loss function in learning non-trivial noise vectors with minimal effect on the output. These findings are supported by a low mean squared error in probability and high prediction consistency.

We advise readers to note the possibility of unboundedness within the loss function's design in Equation (6). While the method presented has empirically shown effective for our initial experiments, we recognize the need for theoretical enhancements. Future work may explore addressing this consideration by adding an explicit constraint on the lower bound of the noise norm to Equation (6). We would also like to note readers that that our main method in Section 3 and main empirical results in Section 4 are based on a different loss function that is free of this concern. We invite readers to comprehensively consider the contribution of our work.

---

### Meta-Review · Area_Chair_6hUp · 2023-12-12

**Metareview:**

The paper presents a new approach to enhancing the adversarial robustness of vision transformers (ViTs) by investigating and utilizing the concept of nullspace. The reviewers commend the paper for its novel idea, clear organization, and the thorough theoretical foundation it provides. The experimental results, particularly in response to the reviewers' feedback, demonstrate the potential of the proposed method in improving robustness against various adversarial attacks. However, the paper also has notable shortcomings. The primary concern is the clarity and justification of using nullspace for adversarial robustness. The application of the approach to CNN architectures yields mixed results, raising questions about its broader applicability. Additionally, some sections of the paper are noted to be confusing in terms of narrative flow and writing style. In essence, while the paper introduces an intriguing concept and shows promising results in specific scenarios, it falls short in presentation as well as convincingly arguing for the broader impact and applicability of its approach.

**Justification For Why Not Higher Score:**

Some limitations pointed out by the reviewers

**Justification For Why Not Lower Score:**

N/A

---

### Decision · Program_Chairs · 2024-01-16

Reject